# LaVi: Efficient Large Vision-Language Models via Internal Feature Modulation

## Abstract

Despite the impressive advancements of Large Vision-Language Models (LVLMs), existing approaches suffer from a fundamental bottleneck: inefficient visual-language integration. Current methods either disrupt the model's inherent structure or introduce long-context computational burdens, severely limiting scalability and efficiency. In this paper, we rethink multimodal integration and present LaVi, a novel LVLM that enables seamless and efficient vision-language fusion through internal feature modulation within the Large Language Models (LLMs). Unlike dominant LVLMs that rely on visual token concatenation, LaVi sidesteps long-context expansion by injecting vision-conditioned *deltas* into the affine parameters of LayerNorm, a ubiquitous component in modern LLMs. This lightweight transformation makes visual input directly modulate the linguistic hidden states, grounding the next-token probabilities in visual evidence. LaVi achieves precise vision–language alignment while retaining the linguistic priors and substantially reducing computation. Across 18 benchmarks covering images, video, and language, LaVi delivers superior or comparable performance with substantial efficiency gains. In addition, it preserves strong linguistic capability. Compared to LLaVA-OV-7B, it reduces FLOPs by 94.0%, accelerates inference by 3.1×, and halves memory consumption. These properties make LaVi a scalable and practical framework for real-time multimodal reasoning. Code and models will be released.

## 1 Introduction

Recently, significant advancements in Large Language Models (LLMs) (Radford et al., 2019; Achiam et al., 2023; Yang et al., 2024; Touvron et al., 2023) have catalyzed the emergence of Large Vision-Language Models (LVLMs) (Bai et al., 2023; Liu et al., 2023a; Awadalla et al., 2023; Tong et al., 2024), demonstrating remarkable capabilities in visual perception and cognitive reasoning (Liu et al., 2024c; Fu et al., 2023; Singh et al., 2019; Hudson & Manning, 2019). While considerable progress has been achieved separately in visual encoding and language generation, the pivotal challenge of effectively integrating visual information into LLMs still remains open.

Existing integration techniques generally fall into two categories. The first, termed ***architectural injection*** (*e.g.*, Flamingo (Awadalla et al., 2023)), augments the original LLMs by introducing additional layers (Alayrac et al., 2022; Meta, 2024; Ye et al., 2024a), such as cross-attention and feed-forward layers, strategically throughout the model. While these modules explicitly insert visual features into the linguistic processing pathway, their introduction inherently disrupts the architectural coherence and processing flow of the original LLMs. Consequently, it can degrade the delicate pre-trained language understanding, risking losing the rich linguistic priors encoded within LLMs (Zhang et al., 2024b; Luo et al., 2024; Wang et al., 2025). The second and currently predominant approach, ***in-context injection*** (*e.g.*, the LLaVA series (Liu et al., 2024b; 2023a; Li et al., 2024a)), integrates visual information by concatenating vision-derived token sequences directly into textual input, treating them as part of the initial context for the LLMs. While preserving architectural integrity, this method introduces significant practical challenges. Specifically, the large number of visual tokens required (*e.g.*, 576 tokens for a single image using CLIP ViT-L/336px (Radford et al., 2021)) leads to severe computational overhead due to the quadratic complexity inherent in self-attention mechanisms (Vaswani et al., 2017). This complexity escalates dramatically when processing high-resolution images or long video sequences, resulting in substantial inference latency and computational bottlenecks, thus hindering real-time applicability.

Through analyzing these methods, we argue that an ideal visual-language integration strategy must satisfy two fundamental principles: 1) ***minimal structural interference***, which ensures the preservation of pretrained linguistic knowledge to support coherent text generation and empower vision-grounded understanding and reasoning; and 2) ***computational scalability***, which mitigates inefficiencies arising from quadratic complexity when processing extensive visual tokens.

Guided by these principles, we propose a new vision–language integration strategy for LVLMs: internal ***Feature Modulation Injection*** (FMI) within the LLMs. At the core of FMI is LayerNorm (LN) (Ba, 2016; Zhang & Sennrich, 2019), a ubiquitous component in modern LLMs that applies learnable affine transformations to rescale and shift hidden states, offering a natural pathway for internal modulation via additive and multiplicative adjustments. Inspired by this, we introduce Vision-Infused Layer Normalization (ViLN), a lightweight extension of standard LN that incorporates visual context into language modeling. Visual features from the vision encoder are transformed by a conditioning module into vision-conditioned *deltas*, which act as residual updates to the original affine parameters of LN. This delta-based modulation refines the normalization in a vision-aware manner, adapting hidden states to the visual context. The adapted hidden states are then passed to the language modeling head to generate next-token predictions, enabling ViLN to ground language generation in vision, akin to existing vision-language integration techniques. Through zero-initialized *deltas*, FMI introduces minimal intervention to the pretrained LLM, leaving its architectural structure and processing flow intact. This design preserves the linguistic priors and relieves the impact on linguistic performance. Moreover, by avoiding visual-token concatenation, it circumvents the quadratic complexity issue, achieving superior computational scalability and efficiently accommodating visual data such as high-resolution images and long videos.

Building on this strategy, we present **LaVi** (Language and Vision Integrator), a novel LVLM that integrates FMI by selectively replacing standard LN with ViLN modules. To provide vision-conditioned modulation, LaVi employs a conditioning module that generates a dedicated visual condition for each text token, enabling fine-grained token-wise alignment. The design of this conditioning module is highly flexible. We explore three alternative implementations: MLP-based, convolution-based, and attention-based approaches, each offering a favorable trade-off between computational efficiency and multimodal performance. The resulting visual conditions are then mapped into token-wise *deltas* through a lightweight projection and injected into the affine parameters of ViLN, thereby modulating the internal linguistic representations in a vision-aware manner.

Benefiting from a significantly reduced context length and a lightweight yet effective visual-language integration strategy, LaVi strikes an impressive balance between computational efficiency and benchmark performance. Comprehensive evaluations across 9 image-based and 6 video-based understanding benchmarks demonstrate that LaVi achieves superior performance comparable to LLaVA-style models while substantially reducing computational overhead. Moreover, it maintains superior linguistic capabilities compared to using other injection strategies. As illustrated in Figure 1, compared to the baseline LLaVA-OV-7B (Li et al., 2024a), LaVi, despite maintaining the same 7B parameter scale, demonstrates substantial improvements in both efficiency and performance. It achieves an impressive 94.0% reduction in FLOPs, operates 3.1× faster, lowers memory consumption by 51.5%, and reduces inference latency from 612.5 ms to just 198.1 ms. Remarkably, LaVi requires even fewer FLOPs than

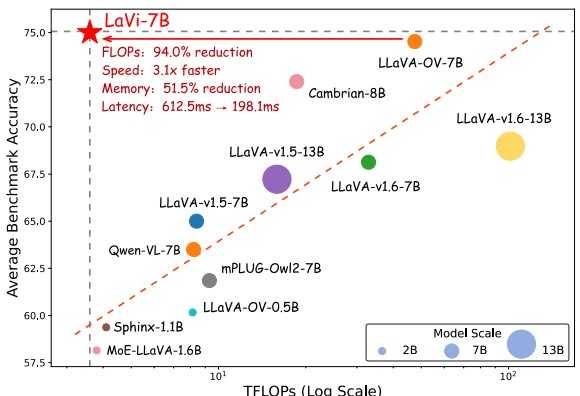

Figure 1: **Comparison between LaVi and open-source LVLMs on image understanding benchmarks.** We report the average accuracy on MM-Bench (Liu et al., 2024c), MME (Fu et al., 2023), TextVQA (Singh et al., 2019), and GQA (Hudson & Manning, 2019). For MME, scores are normalized to percentages. The red dashed line represents the linear fit to all models except LaVi.

LLaVA-OV-0.5B (Li et al., 2024a), yet surpasses it by +15.5 points in benchmark accuracy. These

advancements significantly enhance real-time multimodal interactions, positioning LaVi as a highly efficient alternative in the evolving landscape of LVLMs.

Our contribution can be concluded as:

- We introduce a novel internal feature modulation injection paradigm for LVLMs. It ensures minimal structural interference, effectively preserves pretrained linguistic priors, and achieves computational scalability by avoiding excessive context length expansion.

- We propose LaVi, a highly efficient LVLM capable of comprehensive image and video understanding. LaVi integrates ViLN to achieve fine-grained visual-linguistic alignment and investigates various visual conditioning mechanisms, effectively balancing multimodal performance with computational efficiency.

- LaVi outperforms or matches LLaVA-style baselines across multimodal benchmarks while significantly improving computational efficiency. Compared to the LLaVA-OneVision-7B, LaVi reduces FLOPs by 94.0%, offering an efficient and practical solution for real-time multimodal processing with significantly reduced resource demands.

## 2 RELATED WORK

**Large Vision-Language Models.** Large Vision-Language Models (LVLMs) have significantly advanced multimodal understanding, enabling the integration of vision and language. Closed-source models such as Claude (Anthropic, 2024), GPT (Achiam et al., 2023), and Gemini (Team et al., 2023) series exhibit strong multimodal capabilities. Meanwhile, open-source models like LLaVA (Liu et al., 2023b;a; 2024b; Li et al., 2024a), BLIP (Li et al., 2022; 2023a), Qwen-VL (Bai et al., 2023; Yang et al., 2024), and InternVL (Chen et al., 2024d;c) series have contributed significantly to the community by providing accessible and adaptable alternatives. Recent research has focused on improving input resolution (Liu et al., 2024b; Guo et al., 2024), enhancing training and inference efficiency (Chen et al., 2024a; Wan et al., 2024; Ye et al., 2025), and extending multimodal capabilities to temporal video sequences (Li et al., 2023b; Zhang et al., 2023; Maaz et al., 2023), cognitive alignment (Zhao et al., 2025), various integration (Luo et al., 2025) or reasoning (Xu et al., 2025) approaches.

**Layer Normalization.** Layer Normalization (LN) (Ba, 2016) is a cornerstone of modern Transformers and is widely adopted in LLMs for stabilizing training and regulating hidden state distributions. Its learnable affine parameters provide a natural mechanism for controlling how information flows through attention and feed-forward layers. As models scale, several LN variants (Xiong et al., 2020; Zhang & Sennrich, 2019; Shleifer et al., 2021) have been proposed to better regulate information flow in deep architectures. Building on this controllability, some prior works have explored conditioning affine parameters on external signals, *e.g.*, style (Dumoulin et al., 2016; Ghiasi et al., 2017) or class tags (Brock et al., 2018; Peebles & Xie, 2023), to control the visual appearance of generated images toward a specified style or semantic category. However, existing methods are primarily limited to image synthesis, applying a global conditioning shared across all tokens. To the best of our knowledge, LaVi is the first to extend the LN-based modulation paradigm to LVLMs for cross-modal interaction. To support finer-grained alignment, it introduces a novel token-wise conditioning scheme that generates customized visual *deltas* for each language token.

## 3 METHODOLOGY

### 3.1 PRELIMINARIES

In this section, we begin with a concise overview of the predominant visual-language integration strategies employed in LVLMs. Specifically, existing methods primarily fall into two categories: architectural injection and in-context injection:

**Architectural Injection.** As illustrated in Figure 2a, this approach integrates visual information by inserting additional interaction layers (*e.g.*, cross attention (Awadalla et al., 2023; Alayrac et al., 2022) and hyper attention (Ye et al., 2024a)), enabling fusion between the text sequence $t$ and visual features $v$ within the $\Theta$-parameterized LLM:

$$\mathbf{H}_0 = t, \quad \mathbf{H}_{\ell+1} = \Theta_\ell\big(\Phi_\ell(v, \mathbf{H}_\ell)\big) \tag{1}$$

where $\mathbf{H}_\ell$ denotes the hidden states at layer $\ell$, $\Phi_\ell(\cdot, \cdot)$ represents the inserted cross-modal interaction module, and $\Theta_\ell(\cdot)$ denotes the $\ell$-th layer of the LLM. While this method ensures direct multimodal

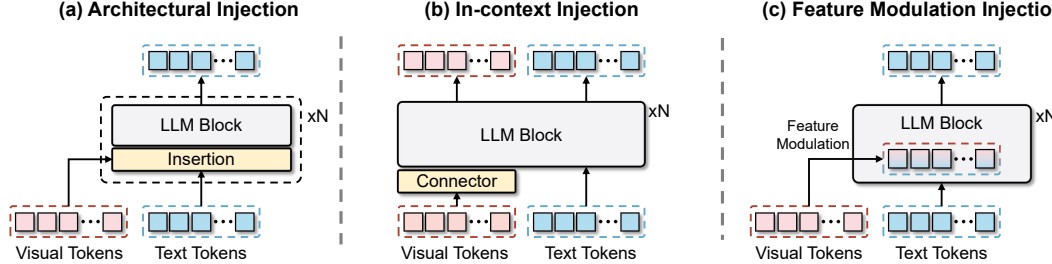

Figure 2: **Comparisons of various vision integration techniques for LVLMs.** (a) Architectural injection: additional layers are inserted into LLM for cross-modal interaction; (b) In-context injection: visual tokens are concatenated before the text sequence as the initial context; (c) Feature modulation injection (Ours): the internal hidden states are modulated by the vision-guided affine transformation.

alignment, it comes at the cost of architectural disruption, requiring extensive modifications to the pretrained LLMs. Such modifications compromise the model's linguistic priors, potentially degrading the generative capabilities in both multimodal and language-only contexts.

**In-context Injection.** As illustrated in Figure 2b, this approach involves mapping visual features $v$ into the LLM's semantic space via a vision-language connector (Li et al., 2023a; 2024a; Liu et al., 2024b; 2023a) and appending them as a visual prefix before the text sequence $t$:

$$\mathbf{H}_0 = [\boldsymbol{v}; \boldsymbol{t}], \quad \mathbf{H}_{\ell+1} = \Theta_\ell(\mathbf{H}_\ell) \tag{2}$$

This method allows cross-modal interaction to occur within the LLM's existing self-attention layers, avoiding explicit structural modifications. However, because self-attention scales quadratically with sequence length (Vaswani et al., 2017), the introduction of numerous visual tokens leads to severe computational inefficiencies. This becomes particularly problematic when processing high-resolution images or long video sequences, where the number of visual tokens grows significantly.

To address the limitations of these approaches, we propose feature modulation injection (FMI), as depicted in Figure 2c. Instead of injecting additional layers or expanding sequence length, FMI incorporates visual information directly into the internal hidden states of the LLM via a lightweight modulation mechanism. More details are provided in the following section.

## 3.2 FEATURE MODULATION INJECTION.

At the core of FMI is the Layer Normalization (LN) module (Ba, 2016; Zhang & Sennrich, 2019), a ubiquitous and essential component in virtually all mainstream LLM architectures. Given an input text sequence $\boldsymbol{t} = \{\boldsymbol{t}_i\}_{i=1}^T$, a typical LLM block processes $\boldsymbol{t}$ as follows:

$$\boldsymbol{t} \leftarrow \boldsymbol{t} + \mathcal{F}_{att}(\text{LN}_1(\boldsymbol{t})) \tag{3}$$

$$\boldsymbol{t} \leftarrow \boldsymbol{t} + \mathcal{F}_{ffn}(\text{LN}_2(\boldsymbol{t})) \tag{4}$$

Here, $\mathcal{F}_{att}$ and $\mathcal{F}_{ffn}$ denote the self-attention and feed-forward sub-layers, respectively. The LN module normalizes the input features via:

$$\text{LN}(\boldsymbol{t}) = \alpha \odot \frac{\boldsymbol{t} - \mu}{\sigma} + \beta = \alpha \odot \hat{\boldsymbol{t}} + \beta \tag{5}$$

where $\mu$ and $\sigma$ are the mean and standard deviation of $\boldsymbol{t}$, and $\alpha$, $\beta$ are learnable affine parameters that control the scaling and shifting of the normalized features. Inspired by this structure, we propose to link the learning of affine parameters to visual features, thereby allowing the visual context to directly influence the hidden states that govern the language modeling distribution. Specifically, we define the following Vision-Infused Layer Normalization (ViLN):

$$\text{ViLN}(\boldsymbol{t}, \boldsymbol{v}) = (\alpha + \Delta\alpha_{\boldsymbol{v}}) \odot \hat{\boldsymbol{t}} + (\beta + \Delta\beta_{\boldsymbol{v}}), \tag{6}$$

Here, $\Delta\alpha_{\boldsymbol{v}}$ and $\Delta\beta_{\boldsymbol{v}}$ are vision-conditioned *deltas* that adaptively adjust the original affine parameters $\alpha$ and $\beta$ in LLM based on visual context. They are dynamically regressed from visual features $\boldsymbol{v}$ through a token-wise conditioning module, which will be detailed later.

Overall, FMI transforms visual information into affine parameters that directly adjust the LLM's internal hidden states via *multiplicative* and *additive* operations. This enables a direct and efficient fusion of vision information at the feature level, eliminating the need for lengthy visual token sequences or additional cross-modal interaction modules.

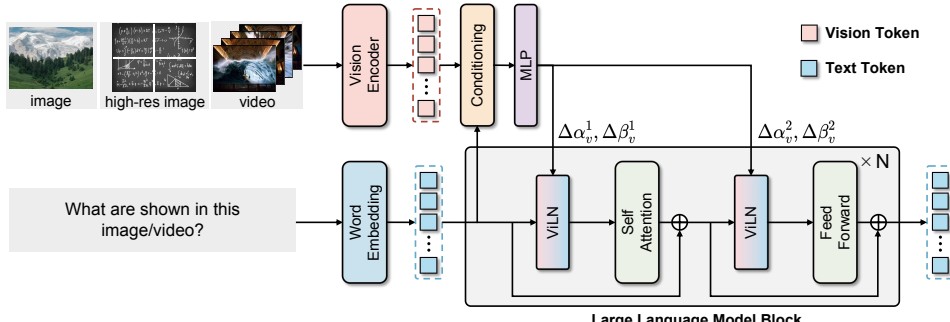

Figure 3: **An illustrative diagram of the overall model architecture.** For a LLM block equipped with ViLN, visual and textual features are fed into the conditioning module to obtain token-wise visual conditions. Through a lightweight MLP, these conditions are then transformed into scale ($\Delta\alpha_{\boldsymbol{v}}$) and shift ($\Delta\beta_{\boldsymbol{v}}$) parameters, which modulate the internal language features of the LLM.

## 3.3 LaVi: A Highly Efficient LVLM

The overall architecture of LaVi is illustrated in Figure 3. After replacing the internal LN of the LLM with ViLN, LaVi leverages a conditioning module to generate token-wise affine parameters *deltas* from visual features $\boldsymbol{v}$, comprising two sets that are applied before the self-attention and feed-forward sublayers, respectively:

$$[\Delta\alpha_{\boldsymbol{v}}^1, \Delta\beta_{\boldsymbol{v}}^1, \Delta\alpha_{\boldsymbol{v}}^2, \Delta\beta_{\boldsymbol{v}}^2] = \text{Swi}\big(\text{Cond}(\boldsymbol{t}, \boldsymbol{v})\big)\mathbf{W} + \mathbf{b} \tag{7}$$

Here, $\text{Swi}(\cdot)$ denotes the Swish activation function (Ramachandran et al., 2017), while $\mathbf{W}$ and $\mathbf{b}$ are learnable projection weights and bias, respectively. This projection is zero-initialized to ensure that the vision-conditioned *deltas* are initially zero, so that the forward pass exactly replicates the original LLM behavior—thereby facilitating stable adaptation and linguistic priors preservation during early training. The conditioning function $\text{Cond}(\cdot)$ is responsible for aggregating visual context relevant to each token in the text sequence $\boldsymbol{t}$. The design of this function is highly flexible. In this paper, we explore three alternative instantiations:

**MLP-based Conditioning**. Inspired by MLP-Mixer (Tolstikhin et al., 2021), we design two sequential MLPs to aggregate visual context. Given a text token $t_i$, we concatenate it with visual features $\boldsymbol{v}$, then transpose the sequence $[t_i; \boldsymbol{v}]$ to interchange token and channel dimensions. A token-mixing MLP integrates information across tokens, and after transposing back, a channel-mixing MLP blends features across dimensions. Vision-aware embedding for $t_i$ is extracted at original position:

$$\text{Cond}_{mlp}(t_i, \boldsymbol{v}) = \Big[\mathbf{MLP}_{channel}\big((\mathbf{MLP}_{token}([\,t_i; \boldsymbol{v}\,]^\top))^\top\big)\Big]_{t_i} \tag{8}$$

**Conv-based Conditioning**. Inspired by ConvMixer (Trockman & Kolter, 2022), we treat the concatenated sequence $[t_i; \boldsymbol{v}]$ as a 1-D signal along the token dimension. We first apply a depth-wise convolution followed by an activation $\sigma$ to mix information between $t_i$ and visual features $\boldsymbol{v}$. Subsequently, a point-wise convolution is utilized to integrate these features across the embedding dimension. The resulting representation at the token position corresponding to $t_i$ provides the vision-aware embedding:

$$\text{Cond}_{conv}(t_i, \boldsymbol{v}) = \Big[\mathbf{Conv}_{point}\big(\sigma(\mathbf{Conv}_{depth}([\,t_i; \boldsymbol{v}\,]))\big)\Big]_{t_i} \tag{9}$$

**Attention-based Conditioning.** We introduce a cross-attention module, where text token $t_i$ is used as the query, while visual tokens $\boldsymbol{v}$ serve as keys and values. Through the attention mechanism, we directly aggregate relevant visual context to produce the vision-aware representation for $t_i$:

$$\text{Cond}_{attn}(t_i, \boldsymbol{v}) = \text{Attention}(t_i\mathbf{W}_Q, \boldsymbol{v}\mathbf{W}_K, \boldsymbol{v}\mathbf{W}_V) \tag{10}$$

We provide further implementation details for the three paradigms in the Appendix. By default, we adopt the attention-based approach due to its simplicity and effectiveness. In Section 4.3, we provide a comparative analysis of each conditioning function, demonstrating that all approaches provide robust multimodal integration with minimal computational overhead.

Table 1: **Performance on 9 image-based benchmarks,** including VQAv2, GQA, VisWiz, ScienceQA, TextVQA, POPE, $MME^P$, MMBench and $SEED^I$. For $MME^P$, the scores are presented as percentages. Along with efficiency and accuracy, we also report the LLM backbone for each baseline.

| Method | LLM | Efficiency | | Performance | | | | | | | | | |
|---|---|---|---|---|---|---|---|---|---|---|---|---|---|
| | | FLOPs | Latency | $VQA^{v2}$ | GQA | VisWiz | SciQA | $VQA^T$ | POPE | $MME^P$ | MMB | $SEED^I$ | Avg. |
| *Baselines with $\leq$ 2B parameters scale* | | | | | | | | | | | | | |
| MoE-LLaVA (Lin et al., 2024) | StableLM-1.6B | 3.8 | 206.4 | 76.0 | 60.4 | 37.2 | 62.6 | 47.8 | 84.3 | 65.0 | 59.4 | – | – |
| MobileVLM-V2 (Chu et al., 2024) | MLLaMA-1.4B | 4.3 | 214.9 | – | 59.3 | – | 66.7 | 52.1 | 84.3 | 65.1 | 57.7 | – | – |
| SPHINX-tiny (Liu et al., 2024a) | TLLaMA-1.1B | 4.1 | 212.3 | 74.7 | 58.0 | 49.2 | 21.5 | 57.8 | 82.2 | 63.1 | 56.6 | 25.2 | 54.3 |
| LLaVA-OV (Li et al., 2024a) | Qwen2-0.5B | 7.8 | 228.0 | 78.5 | 58.0 | 51.4 | 67.2 | 65.9 | 86.0 | 61.9 | 52.1 | 65.5 | 65.2 |
| *Baselines with $\leq$ 8B parameters scale* | | | | | | | | | | | | | |
| Qwen-VL-Chat (Bai et al., 2023) | Qwen-7B | 8.2 | 239.4 | 78.2 | 57.5 | 38.9 | 68.2 | 61.5 | – | 74.4 | 60.6 | 65.4 | – |
| mPLUG-Owl2 (Ye et al., 2024b) | LLaMA2-7B | 9.3 | 278.6 | 79.4 | 56.1 | 54.5 | 68.7 | 54.3 | – | 72.5 | 64.5 | 57.8 | – |
| Cambrian-1 (Tong et al., 2024) | LLaMA3-8B | 18.6 | 393.7 | – | 64.6 | – | 80.4 | 71.7 | – | 77.4 | 75.9 | 74.7 | – |
| LLaVA-v1.5 (Liu et al., 2023a) | Vicuna-7B | 8.4 | 254.4 | 78.5 | 62.0 | 50.0 | 66.8 | 58.2 | 85.9 | 75.5 | 64.3 | 66.1 | 67.5 |
| LLaVA-v1.6 (Liu et al., 2024b) | Vicuna-7B | 32.9 | 502.4 | 81.8 | 64.2 | 57.6 | 70.1 | 64.9 | 86.5 | 76.0 | 67.4 | 70.2 | 71.0 |
| LLaVA-OV (Li et al., 2024a) | Qwen2-7B | 60.4 | 612.5 | 84.5 | 62.2 | 53.0 | 96.0 | 76.1 | 87.4 | 79.0 | 80.8 | 75.4 | 77.2 |
| *Ours* | | | | | | | | | | | | | |
| LaVi-Image | Vicuna-7B | 0.6 | 110.8 | 79.6 | 63.0 | 52.9 | 67.8 | 58.4 | 86.9 | 75.2 | 64.8 | 67.5 | 68.5 |
| $\Delta$ *compare to* LLaVA-v1.5 | | 7.1% | 43.6% | +1.1 | +1.0 | +2.9 | +1.0 | +0.2 | +1.0 | -0.3 | +0.5 | +1.4 | +1.0 |
| LaVi-Image (HD) | Vicuna-7B | 1.7 | 148.6 | 81.4 | 63.7 | 57.8 | 71.7 | 64.3 | 87.0 | 77.5 | 68.1 | 71.6 | 71.5 |
| $\Delta$ *compare to* LLaVA-v1.6 | | 5.2% | 29.6% | -0.4 | -0.5 | +0.2 | +1.6 | -0.6 | +0.5 | +1.5 | +0.7 | +1.4 | +0.5 |
| LaVi | Qwen2-7B | 3.6 | 198.1 | 84.0 | 65.0 | 53.8 | 95.4 | 77.0 | 87.1 | 80.9 | 79.3 | 76.9 | 77.7 |
| $\Delta$ *compare to* LLaVA-OV | | 6.0% | 32.3% | -0.5 | +2.8 | +0.8 | -0.6 | +0.9 | -0.3 | +1.9 | -1.5 | +1.5 | +0.5 |

**Multiple Visual Input Support.** LaVi flexibly accommodates more complex visual inputs—such as high-resolution images and videos—while requiring only minimal structural modifications, making it broadly applicable across diverse vision-language scenarios. Specifically, for high-resolution images, we adopt a tiling strategy, where the image is divided into non-overlapping tiles compatible with the native input size of the vision encoder. Each tile is independently encoded, and the resulting visual tokens are concatenated along the sequence dimension. For videos, we uniformly sample $k$ frames. Each frame is encoded by the vision encoder and undergoes $2\times2$ adaptive pooling. The resulting frame features are concatenated sequentially, with shared temporal position encoding applied to each frame's tokens to capture temporal dynamics.

## 4 EXPERIMENTS

### 4.1 EXPERIMENTAL SETTINGS

**Implementation Details.** We sequentially train three models to investigate the potential and scalability of the proposed architecture. We begin with LaVi-Image, which mirrors the configuration of LLaVA-v1.5 (Liu et al., 2023a), using the CLIP ViT-L/336px (Radford et al., 2021) as the vision encoder and Vicuna-v1.5-7B (Chiang et al., 2023) as the LLM backbone. For high-resolution scalability, we incorporate a dynamic high-resolution mechanism adopted in LLaVA-v1.6 (Liu et al., 2024b) for fair comparison, resulting in LaVi-Image (HD). Furthermore, to explore the full potential of the proposed approach, we extend it to an advanced version, LaVi, which is capable of handling both image and video understanding. For LaVi, in line with LLaVA-OneVision (Li et al., 2024a), we replace the vision encoder with the SigLIP ViT-SO400M/384px (Zhai et al., 2023) and use Qwen2-7B-Instruct (Yang et al., 2024) as the LLM backbone. For all three variants, we uniformly select 25% of the layers in the LLMs and replace their original LN modules with ViLN, upon which FMI is applied. We adopt the attention-based conditioning as the default method. For video inputs, 32 frames are uniformly sampled. All experiments are conducted on 16 NVIDIA A100 GPUs, with the training hyperparameters detailed in the Appendix.

**Training Data.** (1) Pre-training Datasets. We train all three LaVi variants using publicly available images from CC12M (Changpinyo et al., 2021). Following the pre-processing pipeline outlined in (Radford et al., 2021), we retain only samples with resolutions exceeding $448 \times 448$, resulting in a curated subset of 8M samples. (2) Supervised Fine-tuning Datasets. For LaVi-Image, we leverage the instruction datasets corresponding to LLaVA-v1.5 (Liu et al., 2023a), *i.e.*, LLaVA-665K. For LaVi-Image (HD), we leverage the instruction datasets corresponding to LLaVA-v1.6 (Liu et al., 2024b), *i.e.*, LLaVA-760K. For LaVi, we leverage the instruction data from LLaVA-OneVision (Li et al., 2024a). For further details, please refer to the Appendix.

**Evaluation Benchmarks and Metrics.** We evaluate LaVi on both image and video understanding tasks, including 9 image benchmarks and 6 video benchmarks. For evaluation metrics, we report two

Table 2: **Performance on 6 video-based benchmarks,** including EgoSchema, MLVU, VideoMME, MVBench, CinePile and Video-ChatGPT. Along with computational efficiency and accuracy metrics, we also report the number of sampled frames for each video.

| Method | #Frames | Efficiency | | Performance | | | | | |
|---|---|---|---|---|---|---|---|---|---|
| | | FLOPs | Latency | EgoSchema | MLVU | VideoMME | MVBench | CinePile | Video-ChatGPT |
| Video-LLaVA (Lin et al., 2023) | 8 | 32.6 | 488.6 | 38.4 | 47.3 | 39.9 | 43.1 | 25.7 | 2.84 |
| ShareGPT4Video (Chen et al., 2024b) | 16 | 39.2 | 502.7 | – | 46.4 | 43.6 | 51.2 | – | – |
| VideoLLaMA2 (Cheng et al., 2024) | 16 | 27.3 | 465.5 | 51.7 | 48.5 | 46.6 | 54.6 | 44.6 | – |
| LongVA (Zhang et al., 2024a) | 32 | 84.5 | 742.2 | – | – | 51.8 | – | 41.0 | 3.17 |
| LLaVA-NeXT-Video (Liu et al., 2024b) | 32 | 89.6 | 775.4 | 43.9 | – | 33.7 | 46.5 | – | – |
| LLaVA-OV (Li et al., 2024a) | 32 | 129.6 | 1215.6 | 60.1 | 64.7 | 58.2 | – | 49.3 | 3.49 |
| LLaMA-VID (Li et al., 2024d) | 1fps | 182.1 | 2174.3 | 38.5 | 33.2 | 25.9 | 41.9 | – | 2.88 |
| LaVi (Ours) | 8 | 4.2 | 217.0 | 51.8 | 54.2 | 49.4 | 51.8 | 45.6 | 3.03 |
| LaVi (Ours) | 16 | 8.9 | 272.3 | 55.5 | 58.5 | 54.0 | 54.3 | 50.3 | 3.14 |
| LaVi (Ours) | 32 | 18.6 | 401.5 | 58.4 | 62.3 | 57.3 | 56.5 | 54.0 | 3.23 |

categories: *computational efficiency* and *benchmark accuracy*. Specifically, computational efficiency includes FLOPs (T) and latency (ms). Further details could be found in the Appendix.

## 4.2 EVALUATION RESULTS

**Image Understanding Evaluation.** We compare LaVi with baseline models across 9 benchmarks to assess its efficiency and performance, with results presented in Table 1. LaVi strikes a remarkable balance between computational efficiency and performance when compared with all baseline models. The three LaVi variants—LaVi-Image, LaVi-Image (HD), and LaVi —are compared against LLaVA-v1.5, LLaVA-v1.6, and LLaVA-OV, respectively. Compared to their counterparts, they achieve reductions of 14.0×, 19.4×, and 16.8× in FLOPs cost. Despite substantial reductions in computational overhead, the three variants achieve 1.0%, 0.5%, and 0.5% average accuracy improvements across all benchmarks, respectively. These results underscore the superior cross-modal interaction efficiency of FMI compared to existing integration strategies. A more comprehensive comparison of the three strategies is provided in Table 4 of Section 4.3 for further reference.

**Video Understanding Evaluation.** We compare LaVi with advanced video baseline models across 6 widely used benchmarks. To conduct a more comprehensive comparative analysis, in addition to the default setting of 32 frames, we further train two versions utilizing 8 and 16 frames, respectively. The results are presented in Table 2. The superiority of LaVi is strikingly clear. It demonstrates significant computational efficiency, achieving a 6× to 7× reduction in FLOPs compared to baseline models with identical frame counts. Notably, the FLOPs required for the 32-frame LaVi are comparable to half of those needed by the 8-frame Video-LLaVA (Lin et al., 2023). LaVi also consistently surpasses or matches the baseline models in performance across all frame configurations. We further provide a thorough computational overhead analysis associated with frame extension in Section 4.4.

## 4.3 ABLATION STUDY

In this section, we conduct a comprehensive ablation study of the proposed method. For all experiments in this section, we adopt SigLIP ViT-SO400M (Zhai et al., 2023) as vision encoder and Qwen2-7B-Instruct (Yang et al., 2024) as LLM backbone. For training data, we uniformly leverage a 4M subset of the pretraining dataset and LLaVA-665K for alignment and SFT, respectively.

**Fair Comparison of Integration Techniques.** Under *same data and backbone settings*, we present a fair comparison of different integration strategies discussed in Figure 2. We comprehensively assesses *efficiency, linguistic and multimodal capabilities across multiple benchmarks*.

(1) For architectural injection, we evaluate two inserted modules: cross-attention (Awadalla et al., 2023) and hyper-attention (Ye et al., 2024a). (2) For in-context injection, we follow the LLaVA series by concatenating visual features, mapped through a connector, into the text sequence as context. (3) For the proposed FMI, we evaluate the three instantiations introduced in Section 3.3.

The results in Table 4 demonstrate that FMI achieves a superior balance between efficiency and performance. Specifically, it surpasses existing paradigms in both training time and inference overhead. As illustrated in Figure 4, the learning curves of the three injection paradigms during pretraining show that FMI achieves significantly faster convergence, requiring only $1/8$ of the training time compared to in-context injection to reach comparable performance. Furthermore, compared to the other two injection strategies, FMI preserves better linguistic proficiency, demonstrating significant advantages on three prevailing language-only benchmarks.

Table 4: **Fair comparison of integration techniques under identical data and backbone settings.**
We present the total training hours (Time), the FLOPs during inference, and the accuracy results on
three language-only benchmarks and four vision-language tasks.

| Architecture | Efficiency | | Language Benchmarks | | | | Vision-Language Benchmarks | | | | |
|---|---|---|---|---|---|---|---|---|---|---|---|
| | Training Time | FLOPs | MMLU | MBPP | MATH | Avg. | $VQA^T$ | GQA | MMB | $SEED^I$ | Avg. |
| Qwen2-7B-Instruct | – | – | 69.3 | 66.2 | 47.8 | 61.1 | – | – | – | – | – |
| *Architectural Injection* | | | | | | | | | | | |
| Cross Attention | 9.8 | 2.5 | 64.8 | 62.4 | 40.8 | 56.0 | 55.8 | 62.4 | 71.6 | 68.0 | 64.5 |
| Hyper Attention | 8.7 | 2.3 | 65.3 | 62.0 | 41.2 | 56.2 | 56.6 | 61.8 | 71.8 | 68.4 | 64.7 |
| *In-context Injection* | | | | | | | | | | | |
| Concat | 22.0 | 11.4 | 66.2 | 63.6 | 42.4 | 57.4 | **59.0** | **63.4** | 72.0 | 69.2 | 65.9 |
| *Feature Modulation Injection* | | | | | | | | | | | |
| MLP-based | **5.8** | **0.8** | **68.4** | **66.0** | **45.2** | **59.9** | 58.4 | 63.0 | 72.1 | 68.6 | 65.5 |
| Conv-based | 6.0 | **0.8** | 67.7 | 65.4 | 44.9 | 59.3 | 58.0 | 62.7 | 72.4 | 67.5 | 65.2 |
| Attention-based | 6.6 | 0.9 | 68.2 | 65.6 | 44.6 | 59.5 | 58.7 | 63.2 | **72.7** | **69.5** | **66.0** |

**Effect of Modulation at Different Sublayers.**
Each LLM layer comprises two sublayers: self-
attention and feed-forward. We first investigate
the impact of applying ViLN at the sublayer
level. Results are detailed in Table 3. Disabling
ViLN from either sublayer results in a perfor-
mance decrement, notably more pronounced
when removed from the self-attention sublayer.

Table 3: **Effect of modulating different sublayers.**
Injecting visual information into both sublayers
yields optimal results.

| Attn | FFN | $VQA^T$ | GQA | MMB | $SEED^I$ | Avg. |
|---|---|---|---|---|---|---|
| ✗ | ✓ | 55.4 | 61.5 | 71.4 | 69.2 | 64.4 |
| ✓ | ✗ | 57.6 | 62.4 | 72.0 | 67.8 | 65.0 |
| ✓ | ✓ | **58.7** | **63.2** | **72.7** | **69.5** | **66.0** |

This observation likely stems from the self-attention sublayer's pivotal role in handling interactions be-
tween tokens, having a more substantial influence on the efficacy of cross-modal interactions.

**Effect of Modulation Parameter.** ViLN intro-
duces two vision-conditioned *deltas*, $\Delta\alpha_v$ and
$\Delta\beta_v$, which apply multiplicative and additive
modulation to the hidden states. We conduct
an ablation study to isolate the effect of each
component, with results shown in Table 5. Both
*deltas* contribute comparably to performance,
highlighting the equal importance of additive
and multiplicative modulation in integrating visual information effectively.

Table 5: **Effect of modulation parameters**. Each
parameter enhances visual information integration
through corresponding operation.

| $\Delta\alpha_v$ | $\Delta\beta_v$ | $VQA^T$ | GQA | MMB | $SEED^I$ | Avg. |
|---|---|---|---|---|---|---|
| ✓ | ✗ | 58.1 | 62.2 | 70.8 | 67.7 | 64.7 |
| ✗ | ✓ | **59.3** | 62.7 | 69.3 | 66.5 | 64.5 |
| ✓ | ✓ | 58.7 | **63.2** | **72.7** | **69.5** | **66.0** |

**Effect of Modulation Pattern.** We then in-
vestigate the modulation pattern of ViLN by its
frequency and location within the LLM. For
frequency, as shown in Table 6, we vary the pro-
portion of layers applying ViLN from 12.5% to
100%, and observe that 25% yields the best aver-
age performance across benchmarks, suggesting
that a moderate frequency is necessary to bal-
ance the influence of textual and visual signals
on the language modeling distribution. Fixing
the frequency at 25%, we then evaluate four
layer selection strategies: shallow (first 25%),
deep (last 25%), middle (central 25%), and uni-

Table 6: **Effect of modulation frequency**. Apply-
ing ViLN at a moderate frequency yields the best
average performance.

| Config | $VQA^T$ | GQA | MMB | $SEED^I$ | Avg. |
|---|---|---|---|---|---|
| 100% | **59.1** | 62.9 | 71.9 | 68.2 | 65.5 |
| 50% | 58.1 | 62.5 | 70.9 | 69.1 | 65.2 |
| 25% | 58.7 | **63.2** | **72.7** | **69.5** | **66.0** |
| 12.5% | 57.6 | 62.2 | 71.5 | 67.0 | 64.6 |
| shallow | 54.7 | 59.4 | 69.2 | 64.5 | 62.0 |
| middle | 56.5 | 61.6 | 71.4 | 67.3 | 64.2 |
| deep | 57.0 | 60.8 | 70.1 | 65.9 | 63.4 |
| uniform | **58.7** | **63.2** | **72.7** | **69.5** | **66.0** |

form. The results show that uniformly distributing ViLN yields better performance, indicating that a
balanced allocation across layers facilitates more effective and stable cross-modal fusion.

## 4.4 VISUALIZATION AND ANALYSIS

**Linguistic Capabilities Preservation.** For the baselines in Table 4, we compute the cosine distance
between their hidden states and those of the base LLM on MMLU to quantify their drift in language
representation encoding. The layer-wise distances are visualized in Figure 5. LaVi exhibits the
highest similarity to the base LLM, which closely aligns with its performance on language-only
benchmarks in Table 4, thereby validating its advantage in preserving linguistic priors.

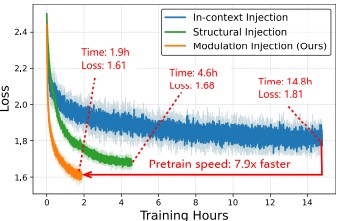 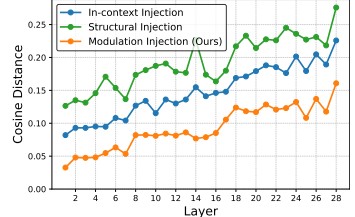 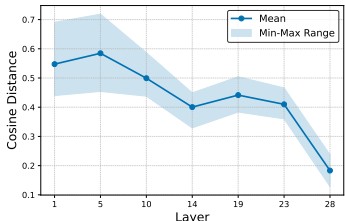

Figure 4: **Training loss of three injection techniques over time.** Our method achieves faster convergence and lower loss.

Figure 5: **Feature distances compared with base LLM.** Our method preserves best linguistic capabilities.

Figure 6: **Features distances before and after ViLN module.** Stronger changes in early layers, while stabilize in deeper layers.

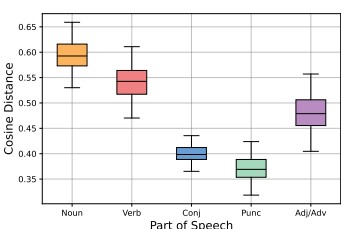 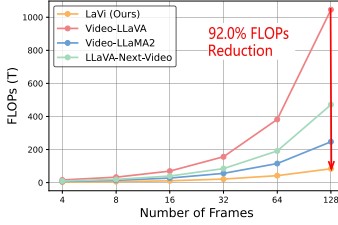 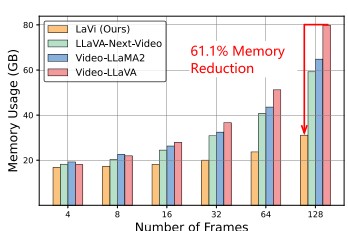

Figure 7: **Modulation influence of LaVi across POS categories.** Semantically rich tokens exhibit stronger modulation influence.

Figure 8: **FLOPs comparison across frame counts.** LaVi achieves significant FLOPs reduction across all frames.

Figure 9: **Memory comparison across frame counts.** LaVi achieves significant Memory reduction across all frames.

**Stronger Modulation Influence on Early Layers.** For LaVi, we then compute the cosine distance between features before and after modulation at each layer as a metric for modulation influence on GQA. Figure 6 shows the average distance (solid line) and the range (shaded area) across tokens. The tokens in early layers undergo significant modulation with notable variance among tokens, while deeper layers show reduced influence and stability. This reflects early layers dynamically establishing cross-modal alignments, while deeper layers refine them into coherent representations.

**Stronger Modulation Influence on Semantically Rich Tokens.** We then evaluate the cosine distances before and after feature modulation across different part-of-speech (POS) categories: nouns, verbs, conjunctions, adjectives/adverbs, and punctuation. POS tagging is performed using NLTK. As illustrated in Figure 7, nouns and verbs exhibit more significant modulation influence compared to conjunctions and punctuation. This is intuitive, as nouns and verbs, which carry richer semantic meaning, are more likely to integrate visual information during cross-modal interactions.

**Superior Vision Sequence Scalability.** High-resolution images and long videos substantially increase visual sequence lengths, resulting in higher computational and memory costs. To evaluate scalability, we compare FLOPs and GPU memory usage of LaVi and existing baselines (Cheng et al., 2024; Lin et al., 2023; Liu et al., 2024b) as the number of frames increases, as shown in Figure 8 and Figure 9. LaVi demonstrates excellent context-length scalability, with both computation and memory overhead growing significantly more slowly than in other models. At 128 frames, it reduces FLOPs and memory usage by 92.0% and 61.1%, respectively, compared to Video-LLaVA, while maintaining superior performance on video understanding benchmarks.

## 5 CONCLUSION

In this work, we propose a novel internal feature modulation injection paradigm for LVLMs, ensuring minimal structural interference and superior computational scalability by avoiding excessive context expansion. Building on this paradigm, we develop LaVi, a highly efficient LVLM that leverages Vision-Infused Layer Normalization (ViLN) for precise visual-linguistic alignment while drastically reducing computational costs. Compared to LLaVA-style models, LaVi achieves 94.0% FLOP reduction, runs 3.1× faster, and significantly lowers latency, establishing LaVi as a highly efficient alternative for vision-language integration.

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

# A  IMPLEMENTATION DETAILS

## A.1  TRIANING DETAILS.

The overall training process adopts a two-stage paradigm, initially involving the pretraining of the conditioning module, followed by instruction tuning. Table 7 and Table 8 presents the details of this two-stage training for LaVi. The implementation includes two sets of vision and LLM combinations: CLIP ViT-L/336px (Radford et al., 2021) + Vicuna (Chiang et al., 2023) or SigLIP ViT-SO400M/384px (Zhai et al., 2023) + Qwen2 (Yang et al., 2024), aligned with the respective LLaVA configurations. Furthermore, consistent with the settings of LLaVA1.6 and LLaVA-OV, we additionally unfroze the ViT during the SFT phase.

## A.2  BENCHMARK DETAILS.

We conduct a comprehensive evaluation of LaVi, including both image and video understanding benchmarks.

**Image-based Benchmarks**  Following the LLaVA framework (Liu et al., 2023a), we conduct experiments across 9 widely recognized benchmarks, including VQA-v2 ($VQA^{v2}$) (Goyal et al., 2017), GQA (Hudson & Manning, 2019), VisWiz (Gurari et al., 2018), ScienceQA-IMG (SciQA) (Lu et al., 2022), TextVQA ($VQA^T$) (Singh et al., 2019), POPE (Li et al., 2023c), MME (Fu et al., 2023), MMBench (MMB) (Liu et al., 2024c), SEED-Bench ($SEED^I$) (Li et al., 2024b). These benchmarks span a broad spectrum of visual tasks. Our evaluation protocols are aligned with those established in the LLaVA framework, ensuring fair consistency.

**Video-based Benchmarks**  We conduct experiments across 6 widely recognized benchmarks, including MVBench (Li et al., 2024c), MLVU (Zhou et al., 2024), EgoSchema (Mangalam et al., 2023), VideoMME (Fu et al., 2024), CinePile (Rawal et al., 2024) and Video-ChatGPT (Maaz et al., 2023). They cover multiple knowledge dimensions and domain focuses, with video durations ranging from a few seconds to several hours.

## A.3  EVALUATION DETAILS.

We adopt LMMs-Eval as our evaluation toolkit. For evaluation prompts, we provide a thorough examination of all evaluation benchmarks utilized in this paper in Table 9. For model efficiency, the FLOPs and latency are calculated using the DeepSpeed toolkit (Team, 2025) on a single A100 GPU without any engineering acceleration techniques.

Table 7: The training details of LaVi based on Vicuna.

| Config | Stage I | Stage II |
|---|---|---|
| LLM backbone | Vicuna-7B | |
| ViT backbone | CLIP ViT-L/336px | |
| Global batch size | 1024 | 256 |
| Batch size per GPU | 64 | 16 |
| Accumulated steps | 1 | 1 |
| DeepSpeed zero stage | 2 | 2 |
| Learning rate | $1\times10^{-3}$ | $2\times10^{-5}$ |
| Learning rate schedule | cosine decay | |
| Warmup ratio | 0.03 | |
| Weight decay | 0 | |
| Epoch | 1 | |
| Optimizer | AdamW | |
| Precision | bf16 | |

Table 8: The training details of LaVi base on Qwen.

| Config | Stage I | Stage II |
|---|---|---|
| LLM backbone | Qwen2-7B | |
| ViT backbone | SigLIP SO400M/384px | |
| Global batch size | 1024 | 256 |
| Batch size per GPU | 64 | 16 |
| Accumulated steps | 1 | 1 |
| DeepSpeed zero stage | 2 | 3 |
| Learning rate | $1\times10^{-3}$ | $1\times10^{-5}$ |
| Learning rate schedule | cosine decay | |
| Warmup ratio | 0.03 | |
| Weight decay | 0 | |
| Epoch | 1 | |
| Optimizer | AdamW | |
| Precision | bf16 | |

# B  CONDITIONING MODULE

To provide a clearer understanding of the proposed conditioning modules, we present PyTorch-style pseudocode implementations for the three vision-conditioned modulation strategies introduced in Section 3.3. Each variant—MLP-based, Conv-based, and Attention-based—is designed to instantiate

Table 9: **Summary of the evaluation benchmarks.** Prompts are mostly borrowed from LMMs-Eval (Bo Li* & Liu, 2024).

| Benchmark | Response formatting prompts |
|---|---|
| POPE (Li et al., 2023c) | – |
| GQA (Hudson & Manning, 2019) | Answer the question using a single word or phrase. |
| VQA$^{v2}$ (Goyal et al., 2017) | Answer the question using a single word or phrase. |
| TextVQA (Singh et al., 2019) | Answer the question using a single word or phrase. |
| MME (Fu et al., 2023) | Answer the question using a single word or phrase. |
| VisWiz (Gurari et al., 2018) | Answer the question using a single word or phrase. When the provided information is insufficient, respond with Unanswerable'. |
| SciQA (Lu et al., 2022) | Answer with the option's letter from the given choices directly. |
| MMBench (Liu et al., 2024c) | Answer with the option's letter from the given choices directly. |
| SEED-Bench (Li et al., 2024b) | Answer with the option's letter from the given choices directly. |
| MLVU (Zhou et al., 2024) | – |
| Video-ChatGPT (Maaz et al., 2023) | – |
| MVBench (Li et al., 2024c) | Only give the best option. |
| VideoMME (Fu et al., 2024) | Answer with the option's letter from the given choices directly. |
| EgoSchema (Mangalam et al., 2023) | Answer with the option's letter from the given choices directly. |
| Cineplie (Rawal et al., 2024) | Answer with the option key (A, B, C, D, E) and nothing else. |

the generic conditioning function $\text{Cond}(\cdot)$ used to derive token-wise affine parameters for Vision-Infused Layer Normalization (ViLN). These modules differ in how they aggregate visual context to influence individual text tokens, yet they all share a common design objective: enabling efficient, token-specific vision-language interaction without altering the LLM's original architecture.

# C ADDITIONAL EXPERIMENTS

## C.1 EVALUATION ON FINE-GRAINED VISUAL UNDERSTANDING

In this section, we further strengthen our evaluation with an assessment of fine-grained visual understanding. Specifically, in addition to the TextVQA (Singh et al., 2019) benchmark provided in the main manuscript, we extend our evaluation to benchmarks such as DocVQA (Mathew et al., 2021), ChartQA (Masry et al., 2022), AI2D (Kembhavi et al., 2016), OCRBench (Liu et al., 2023c), and InfoVQA (Mathew et al., 2022), which require detailed reasoning over figures, documents, and textual content. The results are presented in the Table 10. The results indicate that LaVi performs on par with visual token concatenation in terms of fine-grained visual understanding, thereby validating the effectiveness of our proposed token-wise modulation strategy.

Since fine-grained recognition often relies on representing visual content with a larger number of tokens, we further discuss LaVi's advantages in visual scalability. For traditional approaches, extending the length of visual token sequences comes at a substantial computational cost (*e.g.*, FLOPs increase from 8.4T in LLaVA-v1.5 to 32.9T in LLaVA-v1.6, and further to 60.4T in LLaVA-OV). In contrast, under the same visual token scaling strategy, LaVi's computational cost increases only modestly (*e.g.*, from 0.6T to 1.7T and then to 3.6T). It indicates that LaVi can further enhance the granularity of visual inputs while maintaining low computational overhead. Specifically, we train and then evaluate an extreme case where every input image is divided into a 4×4 grid of tiles for LaVi. The corresponding results are presented in the Table 11. These results demonstrate that LaVi

Table 10: **Performance on 6 fine-grained visual understanding benchmarks.**

| Model | TextVQA | DocVQA | ChartQA | AI2D | InfoVQA | OCRBench | Avg. |
|---|---|---|---|---|---|---|---|
| LLaVA-1.5 | 58.2 | 23.8 | 17.9 | 52.6 | 21.7 | 20.1 | 32.4 |
| LaVi-Image | 58.4 | 24.5 | 17.3 | 52.8 | 21.6 | 21.0 | 32.6 |
| LLaVA-1.6 | 64.9 | 66.9 | 54.2 | 64.6 | 30.2 | 50.3 | 55.2 |
| LaVi-Image (HD) | 64.3 | 66.3 | 55.4 | 65.3 | 31.4 | 51.0 | 55.6 |
| LLaVA-OV | 76.1 | 87.3 | 80.3 | 81.4 | 66.3 | 62.7 | 75.7 |
| LaVi | 77.0 | 87.6 | 81.3 | 80.9 | 67.5 | 63.4 | 76.3 |

Table 11: **Performance on 6 fine-grained visual understanding benchmarks with longer vision token sequence.**

| Model | FLOPs | AnyRes | TextVQA | DocVQA | ChartQA | AI2D | InfoVQA | OCRBench | Avg. |
|-------|-------|--------|---------|--------|---------|------|---------|----------|------|
| LLaVA-OV | 60.4 | Max9 | 76.1 | 87.3 | 80.3 | 81.4 | 66.3 | 62.7 | 75.7 |
| LaVi | 3.6 | Max9 | 77.0 | 87.6 | 81.3 | 80.9 | 67.5 | 63.4 | 76.3 |
| LaVi | 19.5 | 16 | 77.8 | 88.2 | 81.6 | 81.8 | 68.0 | 64.3 | 77.0 |

Table 12: **Performance on 5 multi-image benchmarks.**

| Model | LLaVA-Interleave | MuirBench | Mantis | BLINK | TR-VQA | Avg. |
|-------|------------------|-----------|--------|-------|--------|------|
| GPT-4V (V-Preview) | 60.3 | 62.3 | 62.7 | 51.1 | 54.5 | 58.2 |
| LLaVA-OV | 64.2 | 41.8 | 64.2 | 48.2 | 80.1 | 59.7 |
| LaVi | 65.6 | 43.7 | 63.5 | 46.9 | 81.8 | 60.3 |

offers a significant efficiency advantage when scaling up the visual sequence length to enable more fine-grained understanding of input images.

## C.2 EVALUATION ON MULTI-IMAGE UNDERSTANDING

In this section, we further extend our evaluation on multi-image understanding benchmarks, which represents a crucial frontier for advancing LVLM capabilities.

We begin with a brief introduction to how LaVi performs multi-image understanding. First, analogous to how LaVi distinguishes frames in video inputs using frame embeddings, multi-image inputs are firstly handled by assigning an image-level embedding to all patch tokens belonging to the same image. Distinct embeddings across images allow the conditioning module to differentiate among them. Furthermore, multi-image tasks are typically composed of two basic forms and their combinations. (1) One is joint understanding over multiple images (*e.g.*, describing similarities, differences, or changes across images), where the input typically follows the format $[\text{IMG}_1, ..., \text{IMG}_N, \text{Text}]$, In this case, distinguishing images using the image-level embedding is sufficient for effective conditioning. (2) The other is interleaved image–text understanding (*e.g.*, visual storytelling), where the input may take the form $[\text{IMG}_1, \text{Text}_1, \text{IMG}_2, \text{Text}_2, ...]$. For such settings, we incorporate **causality** into the conditioning module. The tokens in $\text{Text}_i$ are modulated only by the visual features of the preceding images $[\text{IMG}_1, ..., \text{IMG}_i]$. Different images are also distinguished by image-level embedding. Based on these principles, the processing of any multi-image input can be unified as follows: all text segments $\text{Text}_i$ are concatenated and fed into the LLM, while all images $\text{IMG}_i$ are encoded by the ViT and concatenated in their original order. Each token in $\text{Text}_i$ constructs its visual conditioning by aggregating information from all images that precede it in the original sequence, enforced through a causal mask. It allows LaVi to seamlessly support the multi-image training data used in LLaVA-OneVision-Instruct.

We assess the multi-image capability of LaVi on five established multi-image benchmarks, using LMMs-Eval as the evaluation toolkit. The results are summarized in the Table 12. The results indicate that LaVi attains superior or comparable performance to LLaVA-OV-7B, demonstrating the effective support for multi-image understanding.

## C.3 EVALUATION ON VISUAL REASONING TASKS

In this section, we evaluate LaVi on visual reasoning tasks that require complex, multi-step inference. We consider five benchmarks covering mathematical problem solving, visual question answering with multi-hop reasoning, and code-related reasoning. LLaVA-OV-7B (Li et al., 2024a) is used as the main baseline. Following the LMMs-Eval (Bo Li* & Liu, 2024) protocol, we adopt a **reason-first prompt** format, where the model is instructed to explicitly reason before providing the answer. For CoT evaluation, we randomly selected 1k samples from these five benchmarks and provided the images, questions, answers, and the full model outputs to GPT-4o for reasoning quality evaluation (CoT Score). Besides, we further analyze the reasoning depth and CoT consistency. For each model-generated CoT, we prompted GPT-4o to: (i) identify how many distinct reasoning steps are involved in reaching the final conclusion (Reasoning Depth), and (ii) rate the overall logical consistency of the CoT on a scale from 1 to 10 (CoT Consistency).

Table 13: **Performance on Visual Reasoning Tasks.**

| Model | FLOPs | CoT | | | | Benchmark | | | | | |
|---|---|---|---|---|---|---|---|---|---|---|---|
| | | Length | Score | Depth | Consist | MMS | MMV | MathV | AI2D | MMMU | Avg. |
| LLaVA-OV | 60.4T | 132.5 | 7.4 | 4.3 | 8.7 | 62.4 | 57.8 | 63.3 | 81.4 | 48.6 | 62.7 |
| LaVi | 3.6T | 187.6 | 8.0 | 5.0 | 8.4 | 63.5 | 58.6 | 64.2 | 80.9 | 48.8 | 63.2 |

Table 14: **Performance on Caption Generation Tasks.**

| Model | FLOPs | GPT Score | COCO | | NoCaps | |
|---|---|---|---|---|---|---|
| | | | CIDER | BLEU-4 | CIDER | BLEU-4 |
| LLaVA-OV | 60.4T | 8.5 | 137.4 | 41.9 | 86.2 | 34.0 |
| LaVi | 3.6T | 9.0 | 139.7 | 43.3 | 84.8 | 32.6 |

The results in Table 13 show that LaVi achieves comparable or superior performance to LLaVA-OV on visual reasoning tasks, with competitive reasoning depth and CoT consistency. These improvements are achieved while maintaining a significantly lower computational cost. We believe this performance stems from better preservation of language capabilities for LaVi, as the reasoning ability is largely inherited from the base LLM. Additionally, the ability of LaVi to generate in-depth and consistent CoT further demonstrates potential capacity to handle complex multi-step reasoning tasks effectively with futher RL-based tuning.

## C.4 EVALUATION ON CAPTION GENERATION TASKS

In this section, we evaluate LaVi on caption generation tasks, where the goal is to produce meaningful captions for images based on both visual and linguistic understanding. We consider two widely used benchmarks, COCO (Lin et al., 2014) and NoCaps (Agrawal et al., 2019), and use LLaVA-OV-7B (Li et al., 2024a) as the baseline model. Following standard evaluation protocols, we report CIDER (Vedantam et al., 2015) and BLEU (Papineni et al., 2002) to assess the quality of generated captions. Besides, to further evaluate the semantic alignment of the generated captions, we randomly select 1k samples from each benchmark and feed the images along with ground truth captions and model-generated captions into GPT-4o for evaluation. GPT-4o rates the captions on a scale from 1 to 10 (GPT Score), considering aspects such as relevance, coherence, and accuracy in relation to the visual content. The results are summarized in the Table 14.

The results show that LaVi achieves comparable or superior performance to LLaVA-OV on caption generation tasks. This highlights LaVi 's capacity for generating meaningful, contextually relevant captions, reinforcing its efficiency and effectiveness in multimodal tasks.

## C.5 EVALUATION ON PERTURBATION EXPERIMENT

In this section, we assess the robustness of LaVi by evaluating its performance under various perturbations. Specifically, we introduce three types of visual input perturbations, noise, irrelevant images, and adversarial attacks, to simulate potential real-world variations in visual data quality. For each perturbation type, we apply two levels of intensity and measure the resulting performance on 6 standard benchmarks. For adversarial attacks, we apply FGSM-based adversarial perturbations (Goodfellow et al., 2014) to the visual inputs. The attack modifies the image according to the gradient of the loss function, as shown in the following equation:

$$V = V + \epsilon \cdot \text{sign}(\nabla_V J(\theta, V, y)) \quad (11)$$

where $\epsilon$ is the perturbation magnitude, $\nabla_V J(\theta, V, y)$ is the gradient of the loss function with respect to the vision input, and $y$ represents the target label. This perturbation aims to maximize the model's prediction error by pushing the vision input in the direction of the gradient. For noise and irrelevant image perturbations, we add Gaussian noise or related images to the given input:

$$V = V + \sigma \mathcal{N} \quad (12)$$

$$V = V + \lambda \mathcal{V}_{\text{unrelated}} \quad (13)$$

Table 15: Robustness evaluation under different perturbation settings: Gaussian Noise, Unrelated Inputs, and Adversarial Inputs.

| Model | TextVQA | DocVQA | ChartQA | AI2D | InfoVQA | OCRBench | Avg. |
|---|---|---|---|---|---|---|---|
| *Gaussian Noise* | | | | | | | |
| LLaVA-OV-7B | 76.1 | 87.3 | 80.3 | 81.4 | 66.3 | 62.7 | 75.7 |
| $+ \sigma = 0.4$ | 71.7 | 83.8 | 75.7 | 80.2 | 63.0 | 54.9 | 71.6 |
| $+ \sigma = 0.8$ | 65.6 | 78.2 | 67.7 | 72.1 | 56.0 | 53.9 | 65.6 |
| LaVi-7B | 77.0 | 87.6 | 81.3 | 80.9 | 67.5 | 63.4 | 76.3 |
| $+ \sigma = 0.4$ | 73.5 | 85.0 | 77.9 | 80.8 | 65.1 | 57.3 | 73.3 |
| $+ \sigma = 0.8$ | 71.4 | 78.7 | 67.3 | 74.5 | 60.9 | 53.8 | 67.8 |
| *Unrelated Inputs* | | | | | | | |
| LLaVA-OV-7B | 76.1 | 87.3 | 80.3 | 81.4 | 66.3 | 62.7 | 75.7 |
| $+ \lambda = 0.5$ | 74.2 | 86.1 | 78.6 | 80.1 | 65.9 | 60.4 | 74.2 |
| $+ \lambda = 1.0$ | 70.1 | 82.8 | 76.4 | 79.2 | 62.8 | 56.7 | 71.3 |
| LaVi-7B | 77.0 | 87.6 | 81.3 | 80.9 | 67.5 | 63.4 | 76.3 |
| $+ \lambda = 0.5$ | 75.8 | 85.5 | 80.8 | 78.7 | 66.9 | 62.5 | 75.0 |
| $+ \lambda = 1.0$ | 73.6 | 84.1 | 79.3 | 76.6 | 64.4 | 59.2 | 72.9 |
| *Adversarial Inputs* | | | | | | | |
| LLaVA-OV-7B | 76.1 | 87.3 | 80.3 | 81.4 | 66.3 | 62.7 | 75.7 |
| $+ \epsilon = 0.2$ | 70.8 | 84.3 | 73.4 | 78.8 | 62.4 | 56.3 | 71.0 |
| $+ \epsilon = 0.4$ | 68.5 | 79.7 | 72.5 | 77.6 | 59.3 | 52.8 | 68.4 |
| LaVi-7B | 77.0 | 87.6 | 81.3 | 80.9 | 67.5 | 63.4 | 76.3 |
| $+ \epsilon = 0.2$ | 73.2 | 81.3 | 78.6 | 79.4 | 65.8 | 56.5 | 72.5 |
| $+ \epsilon = 0.4$ | 69.6 | 77.8 | 76.4 | 78.5 | 62.3 | 52.1 | 69.5 |

where $\mathcal{N}$ represents a Gaussian noise generated from a standard normal distribution. After applying these perturbations, we compare the performance of LaVi against the baseline LLaVA-OV-7B (Li et al., 2024a) on all 6 benchmarks. The results are summarized in Table 15.

The results show that the proposed modulation mechanism exhibits a reasonable degree of robustness across different types of perturbations, comparable to that of conventional in-context injection methods. Given that visual inputs in LVLM applications rarely contain strong disturbances, we respectfully argue that the robustness of FMI is unlikely to limit its scalability or usability.

# D  CASE STUDY

To provide a more intuitive demonstration of the intrinsic impact of the proposed feature modulation injection paradigm and the capabilities of the novel LVLM LaVi in various scenarios, we present several representative specific examples in this section.

## D.1  IMPACT OF FEATURE MODULATION INJECTION

In this section, to demonstrate the impact of feature modulation injection on the model's output distribution, we conduct the following experiments. First, we input the pure-text question into LaVi and obtain the next-token prediction distribution of the last token. Next, we apply feature modulation using our FMI method, where both the image and the question are simultaneously fed into the model. This results in a modulated next-token prediction distribution, and we present the top three logits for visualization. The results are shown in Figure 10. We observe that the logits distribution for next-token prediction changes before and after the visual feature modulation. Specifically, several interesting observations can be made. As shown in Figure 10 (a), when no visual modulation is applied, the model's prediction of the answer to the question lacks clear distinction and is essentially blind. However, after applying visual modulation, the model's output transitions from this non-targeted distribution to an accurate, targeted one. This illustrates the effect of visual modulation in achieving precise multimodal understanding. As shown in Figure 10 (b), when no visual modulation is applied, the model's prediction shows some language biases, possibly based on previously learned knowledge. After applying visual modulation, the model successfully integrates the visual input and

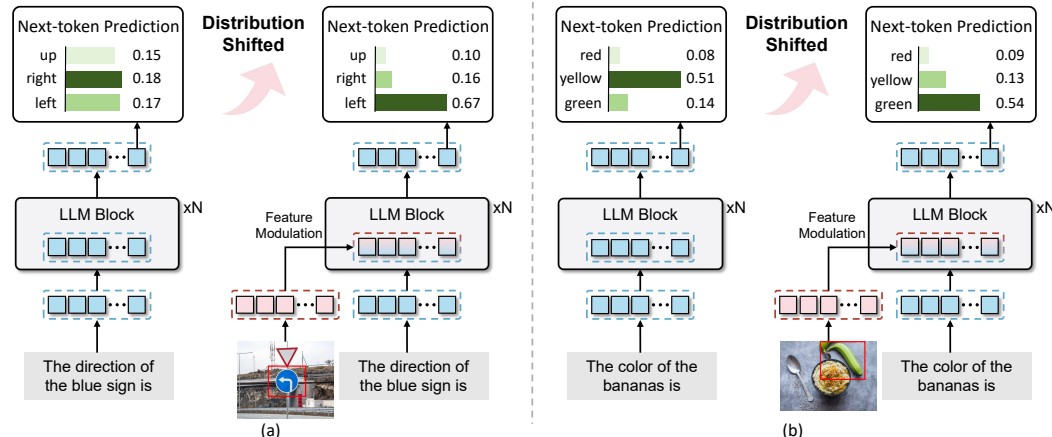

Figure 10: **The change in the logits distribution for next-token prediction before and after the visual feature modulation.**

provides the correct understanding. These examples provide strong evidence for the effectiveness of the feature modulation injection proposed in this work, demonstrating that visual information can directly and effectively influence the feature distribution of the LLM.

## D.2 CASE OF REPRESENTATIVE SCENARIOS

To provide a more intuitive demonstration of the advantages of LaVi as an LVLM compared to existing models, we compare the performance of different models across three representative scenarios: fine-grained visual perception, complex chart reasoning, and long-form video understanding, as shown in Figure 11. The case study highlights LaVi's impressive capabilities in each of these areas. In fine-grained visual perception, LaVi demonstrates its ability to handle intricate visual details with remarkable precision. In the realm of complex chart reasoning, LaVi outperforms previous models such as LLaVA-v1.5 and LLaVA-OV, demonstrating its advanced reasoning skills in interpreting both contextual and numerical data from visual charts. Finally, in the long-form video understanding scenario, LaVi effectively processes and synthesizes extended video content, offering a detailed description of the video. This comprehensive understanding of both visual and contextual information further emphasizes LaVi's strength in managing complex multimodal inputs. Overall, this case study underscores LaVi's superior performance in visual perception, reasoning, and video understanding, highlighting its potential as a powerful tool for multimodal understanding across a wide range of scenarios.

## E USE OF LARGE LANGUAGE MODELS

We used ChatGPT solely as a writing assistant to improve grammar, wording, and LaTeX polishing (*e.g.*, rephrasing sentences for clarity, adjusting tone, and resolving minor formatting issues). The LLM did not contribute to research ideation, problem formulation, methodology design, experiment implementation, data analysis, result interpretation, figure/table creation, or the selection of related work. All technical content, claims, equations, and citations were authored and verified by the authors, who take full responsibility for the paper's contents. Any LLM-suggested edits were treated as copy-editing and were reviewed for accuracy, and no fabricated references were introduced.

## F REPRODUCIBILITY STATEMENT

We have taken concrete steps to facilitate reproducibility. The full model architecture and training objectives are described in Section 3, and the experimental setup is detailed in Section 4. Stage-wise training configurations (optimizer, schedules, precision, batch sizes) are summarized in Table 7 and Table 8 of the appendix, while the evaluation protocols and prompts are also enumerated in Table 9. In addition, the appendix ("Conditioning Module") includes PyTorch-style reference implementations of the modules used in our approach. Together, these materials are intended to enable independent reproduction of our reported results.

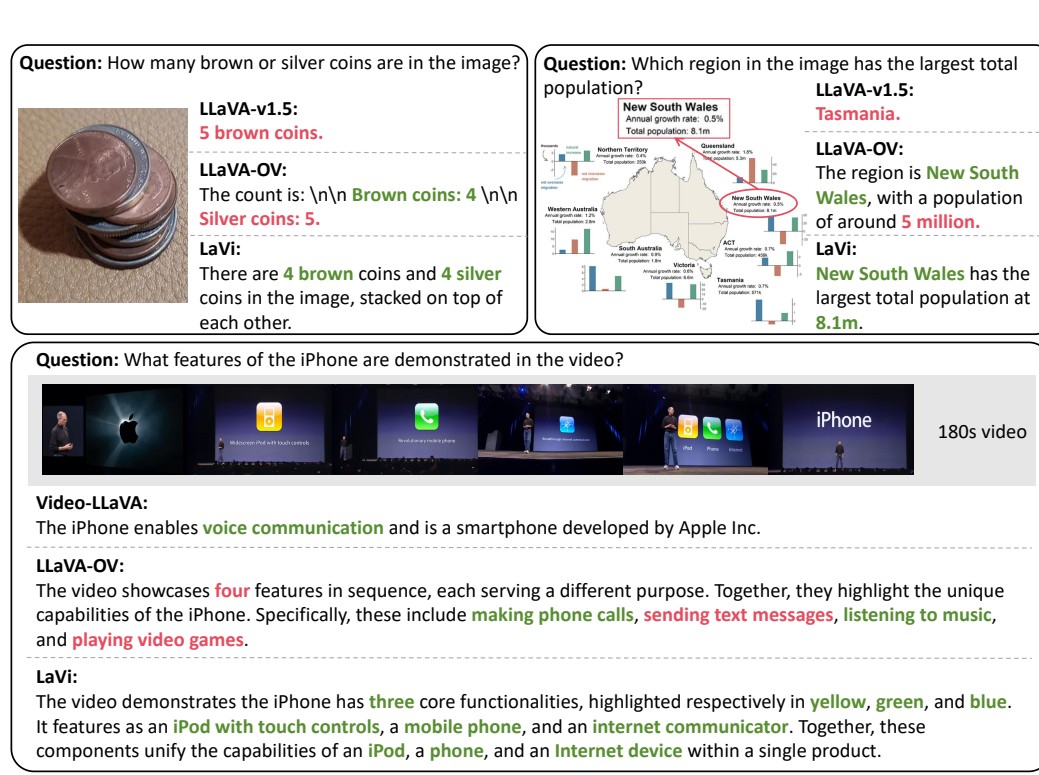

Figure 11: **Case of three representative scenarios: fine-grained visual perception, complex chart reasoning, and long-form video understanding.**

```python
class MLP_Condition(nn.Module):
    def __init__(self, embed_dim: int, num_vis_tok: int,
                 token_exp: int = 4, channel_exp: int = 4):
        super().__init__()
        self.L = num_vis_tok + 1 # total tokens (t_i + v)
        # token-mixing MLP
        self.mlp_token = nn.Sequential(
            nn.Linear(self.L, self.L * token_exp),
            nn.GELU(),
            nn.Linear(self.L * token_exp, self.L))
        # channel-mixing MLP
        self.mlp_channel = nn.Sequential(
            nn.Linear(embed_dim, embed_dim * channel_exp),
            nn.GELU(),
            nn.Linear(embed_dim * channel_exp, embed_dim))

    def forward(self, t_i: torch.Tensor, v: torch.Tensor) -> torch.Tensor:
        assert v.size(1) + 1 == self.L, "Unexpected #visual tokens"
        # concat  (B, L, C) where L = 1 + V
        seq = torch.cat([t_i, v], dim=1) # (B, L, C)
        # Token mixing
        x = seq.transpose(1, 2) # (B, C, L) swap token/chan
        x = self.mlp_token(x)
        x = x.transpose(1, 2) # back to (B, L, C)
        # Channel mixing
        y = self.mlp_channel(y) # (B, L, C)
        # Extract vision-aware embedding for t_i
        return y[:, 0, :]

class Conv_Condition(nn.Module):
    def __init__(self, embed_dim: int, kernel_size: int):
        super().__init__()
        pad = kernel_size // 2
        self.dw = nn.Conv1d(embed_dim, embed_dim, kernel_size,
                            padding=pad, groups=embed_dim)
        self.pw = nn.Conv1d(embed_dim, embed_dim, kernel_size=1)
        self.act = nn.SiLU()

    def forward(self, t_i: torch.Tensor, v: torch.Tensor) -> torch.Tensor:
        # concatenate on token dimension, then transpose for Conv1d
        seq = torch.cat([t_i, v], dim=1).transpose(1, 2) # (B, C, 1+V)
        # depth-wise conv  activation  point-wise conv
        out = self.pw(self.act(self.dw(seq))).transpose(1, 2) # (B, 1+V, C)
        # slice the first token position (corresponding to t_i)
        return out[:, 0, :]

class Attn_Condition(nn.Module):
    def __init__(self, C:int, h:int=8):
        super().__init__()
        self.q = nn.Linear(C, C, False)
        self.k = nn.Linear(C, C, False)
        self.v = nn.Linear(C, C, False)
        self.o = nn.Linear(C, C, False)

    def forward(self, t, v): # t:(B,1,C) v:(B,V,C)
        B = t.size(0)
        def shp(x):
            return x.reshape(B, -1, self.h, self.dk).permute(0, 2, 1, 3)

        q, k, val = map(shp, (self.q(t), self.k(v), self.v(v)))
        attn = (q @ k.transpose(-2, -1)) / math.sqrt(self.dk)
        ctx = (attn.softmax(-1) @ val).transpose(1, 2).reshape(B, 1, -1)
        return self.o(ctx).squeeze(1)
```

Figure 12: Implementation of three conditioning modules in PyTorch.