# OpenReview forum: "LaVi: Efficient Large Vision-Language Models via Internal Feature Modulation"
_ICLR.cc/2026/Conference — Submitted to ICLR 2026_

### Official Review · Reviewer_nXfH · 2025-10-26

**Soundness:** 3
**Presentation:** 4
**Contribution:** 3
**Rating:** 6
**Confidence:** 4

**Summary:**

This paper introduces LaVi, a new framework for building efficient Large Vision-Language Models  through a mechanism called Feature Modulation Injection (FMI), which performs **vision-conditioned modulation** directly within the **LayerNorm** modules of a pretrained LLM. This design achieves minimal architectural disruption and avoids quadratic scaling in sequence length. Experiments show that LaVi attains SOTA performance across 18 benchmarks while reducing FLOPs by up to 94% and inference latency by 3× compared to LLaVA-style models.

**Strengths:**

* **Eelegant integration mechanism**: The proposed **internal modulation** paradigm is conceptually elegant: by modulating existing LayerNorm parameters instead of extending architecture or ccontext length, the model fuses visual information in a manner that is both efficient and minimally intrusive. This is a fresh alternative to the widely used cross-attention or concatenation strategies.
* **Strong empirical results and efficiency**: The experimental results are thorough, spanning 9 image and 6 video benchmarks (Tables 1–2). LaVi consistently matches or exceeds prior models’ accuracy while drastically reducing FLOPs and latency. The quantitative efficiency gains (e.g., 94% FLOP reduction, 3.1× faster inference) are impressive and well-documented.
* **Comprehensive albation studies** (including sublayer modulation, modulation parameters, etc) clearly justify design choices and demonstrate robustness.

**Weaknesses:**

* The paper could benefit from more qualitative examples (e.g., failure cases or visualization of token-level modulation effects) to illustrate how visual context influences language outputs.
* The benchmarks focus on VQA-style metrics; it should be also evaluated on **free-form visual reasoning or caption generation** tasks to confirm broader applicability, as using vision-conditioned deltas to directly modify internal activations maybe very unstable under complex or long-sequence contexts.

**Questions:**

* Have the authors analyzed whether FMI affects the *reasoning depth* or *chain-of-thought consistency* in multimodal tasks? For instance, do FMI models preserve multi-step visual reasoning accuracy compared to in-context injection models?
* Since vision-conditioned deltas directly modify internal activations, what happens when visual inputs are noisy, unrelated, or adversarial? Is the modulation mechanism robust to such perturbations?

---

> ### Author Response · Authors · 2025-11-22
> **Rebuttal by Authors [1/4]**
>
> Dear Reviewer nXfH,
>
> We sincerely appreciate your valuable and insightful comments. We found them extremely helpful for improving our manuscript. We will strive to address each comment in detail, one by one below.
>
> ---
>
> **W1. More qualitative examples**
>
> Thank you for your thoughtful consideration. Qualitative examples are indeed valuable for strengthening the completeness of our paper. We provide a visualization of how **vision modulation influences the next-token prediction distribution** during the LLM decoding process, together with case studies in **three representative scenarios**: fine-grained visual perception, complex chart reasoning, and long-form video understanding. We respectfully invite you to refer to **Appendix D.CASE STUDY** for further details.

---

> ### Author Response · Authors · 2025-11-22
> **Rebuttal by Authors [2/4]**
>
> **W2. Free-form visual reasoning or caption generation**
>
> We greatly appreciate your professional suggestions, and we agree that the two tasks you proposed are both highly valuable. As a result, we have conducted both experiments and report the results below:
>
> * **Free-form visual reasoning:** We evaluated LaVi on five benchmarks that require complex, multi-step reasoning, including tasks such as mathematical problem solving and code generation. We use LLaVA-OV-7B as baseline for comparison.
> Following the LMMs-eval protocol, we employed a **reason-first prompt template**, instructing the model to **explicitly reason before providing the answer**. Additionally, we randomly selected **1k samples** from these 5 benchmarks and sent the images, questions, answers, and the full outputs of the models to GPT-4o for **reasoning quality evaluation (CoT Score)**. GPT-4o rated the reasoning process on a scale from 1 to 10. The results are summarized in the table below:
>
>   | Model | FLOPs| CoT Length | CoT Score | MMStar  | MMVet | MathVista | AI2D | MMMU | Avg. |
>   |-|-|-|-|-|-|-|-|-|-|
>   | LLaVA-OV-7B | 60.4T | 132.45 | 7.44 |  62.4| 57.8 | 63.3 | 81.4 | 48.6 | 62.7|
>   | LaVi-7B | 3.6T| 187.63 | 7.96 | 63.5| 58.6 | 64.2 | 80.9 | 48.8 | 63.2|
>
>   The results show that LaVi achieves performance that is comparable or superior to LLaVA-OV on visual reasoning tasks. We believe it stems from LaVi’s better preservation of language capabilities, as the reasoning ability is mainly inherited from the base LLM.
>
>
> * **Caption generation:** We selected two classic captioning tasks: COCO and Nocaps. We evaluated them using the **CIDER and BLEU** as metrics. Considering that such metrics may not fully capture the semantic quality of generated captions, we additionally sampled **1k instances** from both datasets. We then fed the images, ground truth captions, and the model-generated captions into GPT-4o for evaluation (Caption Score), asking it to rate the caption quality on a scale from 1 to 10. We use LLaVA-OV as the baseline, and the results are summarized below:
>
>   | Model | FLOPs| Caption Score | COCO CIDER  | COCO BLEU4 | NoCaps CIDER | NoCaps BLEU4 |
>   |-|-|-|-|-|-|-|
>   | LLaVA-OV-7B | 60.4T  | 8.49 | 137.4 | 41.9 | 86.2 | 34.0 |
>   | LaVi-7B | 3.6T | 8.96 | 139.7 | 43.3 | 84.8 | 32.6 |
>
>   The results show that LaVi achieves comparable or better performance on the captioning tasks, demonstrating that FMI is capable of handling long-sequence contexts and effectively aligning visual and textual information.

---

> ### Author Response · Authors · 2025-11-22
> **Rebuttal by Authors [3/4]**
>
> **Q1. Reasoning depth & CoT consistency analysis**
>
> We fully understand your question, which naturally follows **Rebuttal by Authors [2/4]**, where we showed that FMI achieves competitive multi-step visual reasoning accuracy. Here, we further evaluate **reasoning depth** and **CoT consistency** on the **same five reasoning benchmarks**. Similar to the CoT score evaluation, we provide GPT-4o with the full context of each sample. Then for each model-generated CoT, we explicitly prompt GPT-4o to: (i) identify how many distinct reasoning steps are involved in reaching the final conclusion (Reasoning Depth), and (ii) rate the overall logical consistency of the entire CoT on a scale from 1 to 10 (CoT Consistency). The aggregated results are summarized below:
>
>   | Model | FLOPs| CoT Length | CoT Score | Reasoning Depth  | CoT Consistency | Accuracy |
>   |-|-|-|-|-|-|-|
>   | LLaVA-OV-7B | 60.4T | 132.45 | 7.44 | 4.25 | 8.73 | 62.7 |
>   | LaVi-7B | 3.6T |187.62 | 7.96 | 4.96 | 8.40 | 63.2|
>
> The results suggest that LaVi preserves multi-step visual reasoning capabilities in terms of reasoning depth and CoT consistency, holding potential for further developing more complex reasoning behaviors through RL-based finetuning.

---

> ### Author Response · Authors · 2025-11-22
> **Rebuttal by Authors [4/4]**
>
> **Q2. Visual inputs perturbations**
>
> This is a very interesting question, and we appreciate your thoughtful consideration. We agree that testing the robustness is indeed meaningful. Specifically, we constructed the test set and obtained the corresponding results as follows:
>
> * **Noisy inputs.** We randomly selected **2k samples** from the evaluated benchmarks. For each image, we added Gaussian noise:
> $$
>  V = V + \sigma \mathcal{N}
> $$
> where $\mathcal{N}$ represents a noise matrix generated from a standard normal distribution and $\sigma$ controls the strength of the noise. The results are summarized below:
>
>   | Model |TextVQA | DocVQA | ChartQA | AI2D | InfoVQA | OCRBench | Avg. |
>   |-|-|-|-|-|-|-|-|
>   | LLaVA-OV-7B |  76.1 | 87.3|	80.3| 81.4 |66.3 | 62.7 | 75.7 |
>   | $\quad$ +$\sigma$=0.4 | 71.7 | 83.8 | 75.7 | 80.2 | 63.0 | 54.9 | 71.6 |
>   | $\quad$ +$\sigma$=0.8 | 65.6 | 78.2 | 67.7 | 72.1 | 56.0 | 53.9 | 65.6 |
>   | LaVi-7B | 77.0 | 87.6 | 81.3 | 80.9 | 67.5 | 63.4 | 76.3 |
>   | $\quad$ +$\sigma$=0.4 | 73.5 | 85.0 | 77.9 | 80.8 | 65.1 | 57.3 | 73.3 |
>   | $\quad$ +$\sigma$=0.8 | 71.4 | 78.7 | 67.3 | 74.5 | 60.9 | 53.8 | 67.8 |
>
>
> * **Unrelated inputs.** When an entirely unrelated image is used as input for questioning, the accuracy is nearly equivalent to **random guessing**. This level of significance makes it difficult to evaluate the robustness of the injection method against perturbations. Therefore, we consider **adding an unrelated image as an interference** on top of the original image:
> $$
>  V = V + \lambda V_{\text{unrelated}}
> $$
> The results are summarized below:
>
>   | Model |TextVQA | DocVQA | ChartQA | AI2D | InfoVQA | OCRBench | Avg. |
>   |-|-|-|-|-|-|-|-|
>   | LLaVA-OV-7B |  76.1 | 87.3|	80.3| 81.4 |66.3 | 62.7 | 75.7 |
>   | $\quad$ + $\lambda$=0.5 |  74.2 | 86.1 | 78.6 | 80.1 | 65.9 | 60.4 | 74.2 |
>   | $\quad$ + $\lambda$=1.0 | 70.1 | 82.8 | 76.4 | 79.2 | 62.8 | 56.7 | 71.3 |
>   | LaVi-7B | 77.0 | 87.6 | 81.3 | 80.9 |67.5 | 63.4 | 76.3 |
>   | $\quad$ + $\lambda$=0.5 | 75.8 | 85.5 | 80.8 | 78.7 | 66.9 | 62.5 | 75.0 |
>   | $\quad$ + $\lambda$=1.0 | 73.6 | 84.1 | 79.3 | 76.6 | 64.4 | 59.2 | 72.9 |
>
>
>
> * **Adversarial inputs.** For each visual input, we apply an adversarial perturbation using the **Fast Gradient Sign Method** (FGSM), as shown in the equation below:
>     $$
>       V = V + \epsilon \cdot \text{sign}(\nabla_V J(\theta, V, y))
>     $$
>     where $\nabla_V J(\theta, V, y)$ is the gradient of the loss function $J$ with respect to the visual input $V$. Here, $\theta$ represents the model parameters and $y$ is the label. The sign function is used to extract the direction of the gradient. This formula represents adding a perturbation to the original visual input **based on the gradient**, ensuring that the perturbation pushes the image in the direction that **maximizes the loss**. The results after applying FGSM are summarized below:
>
>     | Model |TextVQA | DocVQA | ChartQA | AI2D | InfoVQA | OCRBench | Avg. |
>     |-|-|-|-|-|-|-|-|
>     | LLaVA-OV-7B |  76.1 | 87.3|	80.3|	81.4	|66.3 | 62.7 | 75.7 |
>     | $\quad$ + $\epsilon$=0.2 | 70.8 | 84.3 | 73.4 | 78.8 | 62.4 | 56.3 | 71.0 |
>     | $\quad$ + $\epsilon$=0.4 | 68.5 | 79.7 | 72.5 | 77.6 | 59.3 | 52.8 | 68.4 |
>     | LaVi-7B | 77.0 | 87.6 | 81.3 | 80.9 |67.5 | 63.4 | 76.3 |
>     | $\quad$ + $\epsilon$=0.2 | 73.2 | 81.3 | 78.6 | 79.4 | 65.8 | 56.5 | 72.5 |
>     | $\quad$ + $\epsilon$=0.4 | 69.6 | 77.8 | 76.4 | 78.5 | 62.3 | 52.1 | 69.5 |
>
>
> The results show that the proposed modulation mechanism exhibits a reasonable degree of robustness across different types of perturbations, comparable to that of conventional in-context injection methods. Given that visual inputs in LVLM applications rarely contain strong disturbances, we respectfully argue that the robustness of FMI is unlikely to limit its scalability or usability.

---

> ### Comment · Reviewer_nXfH · 2025-11-25
>
> I thank the authors for the detailed response. The manuscript should be revised accordingly.

---

> > ### Author Response · Authors · 2025-11-26
> > **Thank you for your thoughtful and valuable suggestions 😄😄!**
> >
> > Dear Reviewer nXfH,
> >
> > First, we sincerely appreciate your recognition of our response. We have **incorporated each of your valuable suggestions into the revised manuscript**, and we believe these changes have significantly enhanced the completeness and quality of our work.
> >
> > Here, we provide a list of revisions to report how we have incorporated your valuable feedback into the manuscript.
> >
> > * For the **qualitative examples in W1**, we have added them to **Appendix D: Case Study**. This section includes visualizations of how vision modulation influences the next-token prediction distribution, along with three representative examples: fine-grained visual perception, complex chart reasoning, and long-form video understanding.
> >
> > * For the **visual reasoning or captioning benchmark results in W2**, we have supplemented this in **Appendix C.3: Evaluation on Visual Reasoning Tasks** and **Appendix C.4: Evaluation on Caption Generation Tasks**. These sections provide the evaluation protocols, experimental results, and detailed analysis for the 7 additional benchmarks.
> >
> > * For the **CoT reasoning depth and consistency evaluation in Q1**, we have included this analysis in **Appendix C.3: Evaluation on Visual Reasoning Tasks**. Similarly, we provide the evaluation methods, results, and analysis for these two metrics.
> >
> >
> > * For the **perturbation experiment analysis in Q2**, we have added the results in **Appendix C.5: Evaluation on Perturbation Experiment**. This section presents results across three types of perturbations, each with two levels of intensity. These experiments were evaluated on 6 benchmarks, demonstrating the robustness of LaVi.
> >
> > We are truly grateful for the valuable time you dedicated to reviewing our paper and for providing such insightful feedback. If there are any other aspects that you believe require further clarification or additional experiments, we would be more than happy to address them in any way we can. **Your recognition and support mean a great deal to us, and we sincerely hope you might consider reassessing your final rating.** Thank you again for helping us significantly improve the quality of this submission.

---

### Official Review · Reviewer_5Rmb · 2025-10-27

**Soundness:** 2
**Presentation:** 3
**Contribution:** 2
**Rating:** 4
**Confidence:** 4

**Summary:**

This paper introduces LaVi, a Large Vision-Language Model (LVLM) that proposes a new paradigm for vision-language integration by injecting vision-conditioned modulation signals into the affine parameters of LayerNorm modules inside Large Language Models (LLMs). Instead of concatenating visual tokens or modifying model architectures with additional cross-modal layers, LaVi computes visual feature deltas which adaptively modulate internal linguistic hidden states via a Vision-Infused LayerNorm (ViLN) module. The approach aims to minimize structural interference with trained language priors and dramatically reduces computational and memory overhead. Comprehensive experiments are presented across both image and video benchmarks, and the paper analyzes efficiency, performance, and ablation of design choices.

**Strengths:**

1. Innovative Integration Mechanism: The core FMI/ViLN idea—using vision-conditioned deltas to modulate LN affine parameters inside LLMs—is a creative design that shifts the paradigm away from standard token concatenation or explicit cross-modal layers. This is well-motivated in the text (Section 3, Equation for ViLN, Figure 2(c)), and implementation variants (MLP, Conv, Attention) are systematically explored.

2. Quality of Visuals: The architecture diagrams (Figures 2, 3) and efficiency plots (Figures 8, 9) are well-constructed and directly support the claims.

**Weaknesses:**

1. **Weak Positioning to Most Recent and Directly Related Work**: Several highly relevant contemporary works are missing from the related work and comparative analysis (see below), especially those focusing on efficiency (e.g., ATP-LLaVA [1]), cognitive alignment (Beyond Sight [2]), and alternative integration or scaling approaches (Mono-InternVL [3], LLaVA-CoT [4], etc.). This is a significant shortfall given the rapid evolution in the LVLM landscape. The related work (Section 2) is too generic and does not situate LaVi tightly within the latest field advances.

2. **Insufficiently Nuanced Empirical Comparison with SOTA**: While LLaVA, Qwen-VL, and other baselines are included, several top recent efficient or cognitively-aligned LVLMs are absent in Tables 1 and 2—direct apples-to-apples comparisons to works like ATP-LLaVA, Mono-InternVL, or LLaVA-CoT are missing. This makes it difficult to fully contextualize the claimed efficiency/performance trade-offs.

3. **Limited Novelty in Conditioning Mechanisms**: The three conditioning modules (MLP, Conv, Attention) are almost plug-and-play instantiations and, while ablated in Table 4, largely follow established architectural motifs (e.g., Mixer, Convmixer, cross-attention). No clear theoretical or empirical justification is made for why the proposed combination is optimal or uniquely advantageous in the LVLM context.

4. **Sparse Mathematical Elaboration of Modulation Dynamics**: While the ViLN concept is presented with explicit equations (see Equation for ViLN), the actual impact of the modulation on feature space, information flow, and alignment is only superficially discussed. There is no rigorous analysis of, for example, how the injected deltas interact with the primary LN statistics, whether this approach is robust to different LLM sizes, or under what theoretical conditions it preserves or distorts language priors.

5. **Hyperparameter Sensitivity and Practical Implementation Details Under-Explored**: There is only light discussion of the selection/rationale for layer frequency (Table 6) and minimal details about the computational trade-offs across module types (e.g., attention vs. MLP). Further, choices about which layers to modulate and initialization strategies deserve deeper quantitative justification.

6. **Experimental Scope Leaves Out Edge Scenarios**: While the approach is evaluated with different frame counts for video, there is limited evidence for how well it generalizes to much larger LLMs (beyond 7B), or to tasks requiring more complex or compositional visual reasoning. This is especially important given that scalability is a central claim.

[1] Ye, X., Gan, Y., Ge, Y. (2025): ATP-LLaVA: Adaptive Token Pruning for Large Vision Language Models – Focuses on LVLM efficiency, directly related to LaVi’s claims.

[2] Zhao, Y., Yin, Y., Li, L. (2025): Beyond Sight: Towards Cognitive Alignment in LVLM via Enriched Visual Knowledge.

[3] Luo, X., Yang, X., Dou, W. (2025): Mono-InternVL: Pushing the Boundaries of Monolithic Multimodal Large Language Models with Endogenous Visual Pre-training.

[4] Xu, G., Jin, P., Wu, Z. (2025): LLaVA-CoT: Let Vision Language Models Reason Step-by-Step – Enhances multimodal reasoning.

**Questions:**

Please see the weaknesses.

---

> ### Author Response · Authors · 2025-11-22
> **Rebuttal by Authors [1/6]**
>
> Dear Reviewer 5Rmb,
>
> We sincerely appreciate your valuable and insightful comments. We found them extremely helpful for improving our manuscript. We will strive to address each comment in detail, one by one below.
>
> ---
>
> **W1. Strengthening Related Work section**
>
> We appreciate your comprehensive and insightful understanding of the field, and we agree that a broader discussion of recent LVLM developments is highly valuable. Following your suggestion, **we have revised the Related Work section to explicitly incorporate and analyze representative advances** across efficiency-focused methods (e.g., ATP-LLaVA), cognitive alignment approaches (e.g., Beyond Sight), and alternative integration or scaling strategies (e.g., Mono-InternVL, LLaVA-CoT).  We sincerely thank you for this helpful recommendation, which has contributed to improving the completeness and clarity of our manuscript.

---

> ### Author Response · Authors · 2025-11-22
> **Rebuttal by Authors [2/6]**
>
> **W2. Comparison with related works**
>
> Thank you for the thoughtful reminder. As a supplement, we evaluate the efficiency–performance trade-offs of ATP-LLaVA, Mono-InternVL, and LLaVA-CoT, comparing them against LaVi. Before presenting the comparison, we would like to respectfully emphasize that **although these methods are motivated by different goals, they all rely on an in-context injection paradigm to enable multimodal understanding**. In contrast, LaVi aims to fundamentally improve this interaction mechanism by replacing token concatenation with feature-level modulation. Below, we briefly summarize the core ideas of the three works:
>
> * ATP-LLaVA introduces adaptive token pruning to reduce the number of visual tokens fed into the LLM, thereby improving inference efficiency.
>
> * Mono-InternVL proposes a monolithic multimodal model without an external visual encoder, but still integrates visual information through large-scale visual token sequences appended to the LLM input.
>
> * LLaVA-CoT enhances multimodal reasoning by incorporating chain-of-thought supervision into LVLM training while keeping the standard visual-token concatenation pipeline.
>
> We measured accuracy and FLOPs across 5 benchmarks. The related efficiency–performance trade-offs are summarized as follows:
> | Model| Param | FLOPs | MMB | SEED-I | VQAv2 | AI2D | GQA |
> |-|-|-|-|-|-|-|-|
> | ATP-LLaVA | 7B | 3.0T |  66.0 |  57.3 | 76.4| -- | 59.5 |
> | Mono-InternVL | 3B | 4.7T | 65.5 | 67.4 | -- | 68.6 | 59.5 |
> | LLaVA-CoT | 11B | 22.9T | 75.0 | -- | -- |  78.7 | -- |
> | LaVi | 7B | 3.6T| 79.3 | 76.9 | 84.0 | 80.9 | 65.0 |
>
> The results demonstrate that, among efficiency-focused LVLMs with comparable computational budgets, LaVi achieves obviously better performance than **ATP-LLaVA**. Moreover, compared with **LLaVA-CoT (11B)**, LaVi attains comparable or even superior accuracy while using only **~15%** of its FLOPs. These findings indicate that LaVi strikes an excellent balance between performance and efficiency.

---

> ### Author Response · Authors · 2025-11-22
> **Rebuttal by Authors [3/6]**
>
> **W3. Clarification on conditioning mechanism**
>
> Thank you for raising this question. We greatly appreciate the opportunity to clarify this point. We wish to address it from the following three perspectives:
>
> * **The role of the conditioning mechanism.** Our core motivation lies in **introducing a new paradigm for injecting visual information into LLMs**. To the best of our knowledge, Feature Modulation Injection (FMI) is the first approach to perform cross-modal interaction inside the LLM through feature-level modulation. This paradigm eliminates the need to expand the input sequence via visual token concatenation or to insert explicit interaction layers into the LLM. Similar to the fact that connectors in in-context injection (e.g., Linear [1], MLP [2], or QFormer [3]) can take various forms, **FMI does not rely on any specific module architecture**. The conditioning mechanism simply produces the modulation parameters ($\Delta{\alpha_v}, \Delta{\beta_v}$) required to realize this new interaction scheme.
>
> * **Why we explore multiple conditioning modules.** We would like to respectfully clarify that the three conditioning variants were not introduced to determine which one is inherently superior. Rather, our goal was to **demonstrate the generality and robustness of FMI**. As long as a conditioning module can aggregate visual information conditioned on each text token, it can seamlessly operate within the FMI framework.
> The observation that MLP-, Conv-, and Attention-based designs achieve comparable performance and efficiency further indicates that **the effectiveness of FMI is not sensitive to the specific instantiation of the conditioning mechanism**. It reveals that FMI is a **paradigm-level contribution** rather than a module-level modification.
>
> * **Why use attention-based conditioning module by default.** Considering that it achieves slightly higher accuracy under comparable computational cost, and maintains architectural consistency with the underlying LLM, we adopt the attention-based conditioning module as the default choice in our main experiments. We respectfully refer you to **Rebuttal by Authors [5/6]** for further technical details.
>
> [1] Liu H, Li C, Wu Q, et al. Visual instruction tuning[J]. Advances in neural information processing systems, 2023, 36: 34892-34916.
>
> [2] Liu H, Li C, Li Y, et al. Improved baselines with visual instruction tuning[C]//Proceedings of the IEEE/CVF conference on computer vision and pattern recognition. 2024: 26296-26306.
>
> [3] Li J, Li D, Savarese S, et al. Blip-2: Bootstrapping language-image pre-training with frozen image encoders and large language models[C]//International conference on machine learning. PMLR, 2023: 19730-19742.

---

> ### Author Response · Authors · 2025-11-22
> **Rebuttal by Authors [4/6]**
>
> **W4. In-depth analysis of modulation dynamics**
>
> We deeply appreciate your insightful observations. In response to each of your valuable points, we provide quantitative analysis and illustrative cases of the modulation dynamics. If you believe any additional experiments or analyses would be helpful, we are more than happy to conduct and include them.
>
> * **Interaction between injected deltas and primary LN statistics.** The formulation of ViLN can be decomposed into two parts:
> $$
>   \text{ViLN}(t, v) =(\underbrace{\hat{t} \odot \alpha + \beta} _ {\text{LLM Part}}) + (\underbrace{\hat{t} \odot \Delta \alpha_v + \Delta \beta_v } _ {\text{Vision Part}})
> $$
>
>    It shows that the visual information contributes additively as a residual term on top of the original LN output, effectively refining the primary LN statistics (mean $\mu$ and variance $\sigma$). This refinement characterizes the interaction between the injected deltas and primary statistics, which is designed to **pull text features toward regions in feature space that better reflect the visual context.** Conceptually, this objective is consistent with the mainstream in-context injection paradigm. Both approaches ultimately aim to blend visual information into text representations so that the hidden-state evolution of text tokens and the next-token prediction of LLMs are both guided by the visual input. To verify this, we measure the **cosine similarity** between text tokens and visual features on benchmarks reported in Table 1 of the manuscript. Specifically, we compute this similarity before and after ViLN modulation for FMI, and additionally report the similarity before and after the attention operation for in-context injection as a comparison.
>
>     |Paradigm|Before Injection|After Injection|
>     |-|-|-|
>     |in-context | 0.29 ± 0.24 | 0.43 ± 0.29 |
>     | feature modulation| 0.32 ± 0.13 | 0.47 ± 0.17|
>
>     These results indicate that **the interaction indeed moves text representations closer to visual semantics**, and does so to a degree comparable to standard in-context injection, while using a more direct feature-level formulation.
>
>
> * **Scaling with LLM size.** To explore the scalability of our method across different model sizes, we evaluate LaVi using three variants of Qwen2 with different parameter scales: 0.5B, 7B, and 72B. For the 0.5B and 72B models, we follow the same setup as used for LaVi-7B. The image-based performance is summarized as follows:
>
>   | Model| MMB | SEED-I | TextVQA | DocVQA | ChartQA | AI2D | InfoVQA |
>   |-|-|-|-|-|-|-|-|
>   | LaVi-0.5B | 51.6 | 67.4 | 65.9 | 69.5 | 61.2 | 57.4 | 43.1 |
>   | LaVi-7B |  79.3 | 76.9 | 77.0 | 87.6 | 81.3 | 80.9 | 67.5 |
>   | LaVi-72B | 85.7 | 77.8 | 80.5 | 93.8 | 84.7 | 85.7 | 75.6 |
>
>   The video-based performance is summarized as follows:
>   | Model| Frame | EgoSchema | MLVU | VideoMME | MVBench |
>   |-|-|-|-|-|-|
>   | LaVi-0.5B | 32 | 28.7 |  51.6 | 45.3 | 44.9 |
>   | LaVi-7B | 32 | 58.4 | 62.3 |  57.3 | 56.5 | 58.6 |
>   | LaVi-72B | 32 |61.8 | 68.5 | 64.1 | 60.2 |
>
>   **LaVi scales favorably across model sizes**, with consistent performance improvements observed from 0.5B to 72B on both all benchmarks.
>
> * **Preservation of language priors.** We believe that FMI preserves language priors from two perspectives.
>
>   * **Theoretical perspective.** As described in the paper, the vision-conditioned deltas are initialized to zero, ensuring that the model begins exactly from the pretrained LLM behavior and that visual information is injected in a stable and incremental manner. This design follows the same principle as several successful residual adaptation techniques such as LoRA and ControlNet. Furthermore, it allows us to derive a theoretical upper bound via a first-order residual decomposition:
>   $$
>   ||h_{\text{FMI}}(x)-h_{\text{LN}}(x)||_2\le||\hat{x}||_2 \cdot ||\Delta \alpha_v||_2 + ||\Delta \beta_v||_2
>   $$
>   It highlights that FMI modifies the pretrained representation through a linear residual whose effect is limited by the norms of the learned deltas. Since the deltas are zero-initialized and optimized under residual learning dynamics, their magnitude remains very stable in practice, keeping the modulation within a predictable and well-controlled deviation from the original LLM.
>
>   * **Empirical perspective.** As shown in Table 4 of the manuscript, we compare FMI with other injection strategies under same settings. When evaluated on language benchmarks, FMI consistently achieves higher accuracy. Moreover, as illustrated in Figure 5, we feed MMLU samples through models trained with each injection paradigm and compute the layer-wise cosine distance between their hidden states and those of the original LLM. FMI exhibits significantly smaller distances across layers, demonstrating that FMI faithfully preserves the pretrained language priors.

---

> ### Author Response · Authors · 2025-11-22
> **Rebuttal by Authors [5/6]**
>
> **W5. Additional hyperparameter exploration**
>
> Thank you for suggesting these experiments, which are indeed important for improving the completeness of our study. **We have conducted all the additional investigations you pointed out**, and we summarize the corresponding results below.
>
> *  **Computational trade-offs across conditioning modules.** We would first like to respectfully emphasize that **Table 4 of the manuscript already compares models trained with the three conditioning modules** (MLP-, Conv-, and Attention-based). The comparison includes **seven benchmark accuracies, training time, and inference FLOPs**, providing an initial view of their efficiency–performance trade-offs. In addition, Appendix B provides the complete code implementations of all three conditioning modules. To provide a clearer comparison, we report **more efficiency metrics** and additionally include their **average accuracy across the all 9 benchmarks in Table 1** as a reference.
>
>    | Module| Train Time | Train Peak Mem | Infer FLOPs | Infer Latency  | Acc |
>    |-|-|-|-|-|-|
>    | LaVi w/ MLP-based  | 5.84h |  74.7G | 0.83T | 120.7ms | 69.2 |
>    | LaVi w/ Conv-based|  5.95h | 73.6G | 0.76T | 118.5ms | 68.6
>    |LaVi w/ Attention-based| 6.57h | 76.2G | 0.92T |  122.0ms | 70.0 |
>
>    Given that the attention-based conditioning module achieves slightly higher accuracy under comparable computational cost and the architectural consistency with the underlying LLM, we adopt the attention-based design as the default choice in our main experiments.
>
> *  **Which layers to modulate.** We wish to kindly remind that **we have already conducted ablation and analysis regarding which layers should be modulated**, as presented in Table 6. This table not only evaluates different layer frequencies, but also analyzes which specific layers are most effective when the modulation ratio is fixed at 25%. Specifically, we compare four strategies: shallow (first 25%), deep (last 25%), middle (central 25%), and uniform. The results show that the **uniform strategy leads to more effective and stable cross-modal fusion**, providing a clear rationale for our choice in the main experiments.
>
>
> *  **Initialization strategies.** By default, the projections used to generate the vision-conditioned deltas are initialized to zero, ensuring that the model starts exactly from the pretrained LLM behavior and that visual information is injected in a stable and incremental manner. Following your suggestion, we additionally experiment with initializing these projections using a standard Xavier initialization, and train the model under the same settings for a fair comparison. The results on the 3 language benchmarks and 4 multimodal benchmarks are summarized below:
>
>    | Initialization | MMLU | MBPP | MATH | T-Avg. | TextVQA | GQA | MMB | SEED-I |  I-Avg. |
>    |-|-|-|-|-|-|-|-|-|-|
>    | Zero (default) | 68.2 | 65.6 |44.6 |59.5| 58.7 |63.2| 72.7 |69.5 |66.0|
>    | Xavier | 66.9 | 64.1 |42.9 | 58.0 | 57.8 | 62.3 | 71.9 |  70.1| 65.5 |
>
>    The results indicate that zero initialization yields better performance on both language and multimodal tasks, with the greatest benefit observed on language tasks. This can be attributed to the fact that zero initialization allows the model to retain the pretrained language priors without introducing any initial perturbation, providing a more stable foundation for visual information injection.

---

> ### Author Response · Authors · 2025-11-22
> **Rebuttal by Authors [6/6]**
>
> **W6. Experiment on Edge Scenarios**
>
> Thank you for your thoughtful and detailed consideration. We have conducted experiments on **both much larger LLMs and tasks requiring more complex or compositional visual reasoning**, and we provide the results below:
>
> * **Much larger LLMs.**，As discussed in **Rebuttal by Authors [4/6]**, we trained a 72B model and evaluated it on 11 benchmarks, including both video and image-based tasks. The results demonstrate that, compared to the 7B model, the 72B model shows significant improvements across all benchmarks.
>
> * **Complex reasoning tasks.** We selected 5 benchmarks that require complex reasoning and compositional visual understanding, and evaluated LaVi on these tasks. These benchmarks include tasks such as mathematical reasoning and coding problems, which require the model to handle structured reasoning and multi-step problem-solving. As before, we use LLaVA-OV-7B for comparison.
> Following the LMMs-eval protocol, we employed a **reason-first** prompt template, instructing the model to explicitly reason before providing the answer. Additionally, we randomly selected **1k samples** from these 5 benchmarks and sent the images, questions, answers, and the full outputs of the models to GPT-4o for **reasoning quality evaluation (CoT Score)**. GPT-4o rated the reasoning process on a scale from 1 to 10. The results are summarized in the table below:
>
>   | Model | CoT Length | CoT Score | MMStar  | MMVet | MathVista | AI2D | MMMU | Avg. |
>   |-|-|-|-|-|-|-|-|-|
>   | LLaVA-OV-7B | 132.4 | 7.44 |  62.4| 57.8 | 63.3 | 81.4 | 48.6 | 62.7|
>   | LaVi-7B | 187.6 | 7.96 | 63.5| 58.6 | 64.2 | 80.9 | 48.8 | 63.2|
>
>   The results show that LaVi outperforms LLaVA-OV-7B on these tasks, demonstrating  complex and compositional visual reasoning capabilities.

---

> > ### Comment · Reviewer_5Rmb · 2025-11-25
> > **Response to rebuttal**
> >
> > I would like to thank the authors for their effort in preparing the rebuttal. The response largely resolves my initial concerns regarding the efficiency/performance advantages and the theoretical basis for the modulation mechanism. And the scalability analysis of LaVi is now much clearer. Based on these clarifications, I acknowledge the potential of LaVi as an innovative vision-language integration mechanism.  I have raised my score to Accept.

---

> > > ### Author Response · Authors · 2025-11-25
> > >
> > > Dear Reviewer 5Rmb,
> > >
> > > Thank you for your recognition of our work and the time and effort you have invested as a reviewer!
> > >
> > > We will adhere to your valuable suggestions to refine our manuscript accordingly.

---

### Official Review · Reviewer_u2BS · 2025-10-30

**Soundness:** 2
**Presentation:** 2
**Contribution:** 2
**Rating:** 2
**Confidence:** 5

**Summary:**

The paper proposes LaVi, a Large Vision-Language Model (LVLM) designed for computational efficiency. The central idea is to inject visual information into a Large Language Model (LLM) by modulating the affine parameters of its internal LayerNorm (LN) modules. The authors claim that this approach achieves state-of-the-art performance on par with existing models like LLaVA, while significantly reducing FLOPs, latency, and memory usage.

**Strengths:**

1.  **Addresses a Critical Problem:** The paper tackles the highly relevant and important problem of computational inefficiency in Large Vision-Language Models (LVLMs). As models grow in capability and are applied to longer visual contexts (e.g., high-resolution images, videos), the quadratic complexity of self-attention becomes a major bottleneck. The work's focus on creating a more scalable and practical framework is well-motivated and timely.

2.  **Impressive Efficiency Gains:** The primary strength of the proposed method, LaVi, lies in its remarkable computational efficiency. The paper provides compelling evidence (Table 1, Figure 8, Figure 9) that its approach significantly reduces FLOPs, inference latency, and memory consumption compared to widely-used in-context injection methods like LLaVA.

**Weaknesses:**

1.  **Limited Novelty:** The core idea of modulating normalization layer parameters with external conditioning is not new. This concept is well-established in computer vision, most notably with Adaptive Instance Normalization (AdaIN) for style transfer (Dumoulin et al., 2016) and conditional normalization in generative models (Brock et al., 2018). The paper's primary contribution is the application of this existing idea to the domain of LVLMs. While this extension is acknowledged, it can be viewed as a straightforward, incremental adaptation rather than a fundamental architectural innovation. The paper lacks a deep, principled discussion on *why* LN modulation is a superior mechanism for vision-language fusion over other potential pathways, making the approach feel more like an engineering heuristic than a novel scientific paradigm.

2.  **Unconvincing and Inconsistent Performance:** The paper's central claim of achieving "state-of-the-art multimodal performance" is not well-supported by the results. The reported efficiency gains appear to come at the expense of accuracy on several key benchmarks, indicating a classic speed-accuracy trade-off rather than a Pareto improvement.
    *   **Performance Regression on Key Benchmarks:** In Table 1, when comparing "LaVi" against "LLaVA-OV," the average score improvement (+0.5%) obscures significant performance regressions on individual, challenging benchmarks such as VQAv2 (-0.5), ScienceQA (-0.6), and notably MMBench (-1.5). Sacrificing accuracy on established benchmarks for efficiency is a valid trade-off, but it contradicts the claim of achieving SOTA performance.
    *   **Incomplete Benchmark Suite:** The evaluation is missing many contemporary and challenging benchmarks that are crucial for assessing the true capabilities of modern LVLMs. For instance, fine-grained visual understanding benchmarks (e.g., OCRBench, DocVQA, ChartQA) and advanced reasoning benchmarks in the STEM domain (e.g., MMMU, MathVista, MathVerse) are not evaluated. Similarly, for video, important long-video benchmarks like Video-MME-Long and LongVideoBench are absent, which makes it difficult to assess the model's claimed scalability in truly long-context scenarios.

3.  **Insufficient and Outdated Baselines:** The experimental comparison is narrow and heavily focused on the LLaVA family. This presents a skewed and incomplete picture of the current LVLM landscape.
    *   **Missing SOTA Models:** The paper fails to compare against numerous stronger and more recent open-source models, such as the InternVL series, Qwen-VL series (e.g., Qwen2-VL, Qwen2.5-VL). Without these comparisons, the "state-of-the-art" claim is unsubstantiated.
    *   **Missing Diverse Architectures:** The comparison also lacks other relevant architectures that explore efficiency and alternative fusion mechanisms. Models like mPLUG-Owl3, VideoChat-Flash, or CrossLMM should have been included to properly situate LaVi's contribution and demonstrate its superiority over a wider range of architectural designs.

4.  **Insufficient Justification and Superficial Analysis:** The methodological choices lack strong justification, and the analysis is not deep enough to be truly insightful.
    *   The paper introduces three conditioning modules (MLP, Conv, Attention) but offers little intuition for their design or a compelling, principled reason for selecting the attention-based variant beyond a simple empirical comparison.
    *   The analysis in Section 4.4, while visually appealing, is superficial. For example, observing that nouns and verbs receive stronger modulation (Figure 7) is intuitive but unsurprising, and does not in itself validate the overall approach. The analysis demonstrates *that* the model exhibits certain behaviors but falls short of explaining *why* this behavior leads to a more effective or principled form of vision-language integration.

5.  **Misleading Framing of Contribution:** The paper frames its contribution as achieving top performance *and* top efficiency simultaneously. However, the evidence points to a classic engineering trade-off: the model is faster but measurably less accurate on several tasks and is not compared against the true state-of-the-art. This trade-off is valuable, but it should be presented as such. By claiming "state-of-the-art performance," the paper overstates its contribution and sets unrealistic expectations. A more accurate and honest framing would be "A New Efficiency-Accuracy Trade-off for LVLMs," which would be a solid but less groundbreaking contribution.

**Questions:**

See the weakness.

---

> ### Author Response · Authors · 2025-11-22
> **Rebuttal by Authors [1/5]**
>
> Dear Reviewer u2BS,
>
> We sincerely appreciate your valuable and insightful comments. We found them extremely helpful for improving our manuscript. We will strive to address each comment in detail, one by one below.
>
> ---
>
> **W1. Novelty Discussion**
>
> Thank you for your question. We appreciate your broad and in-depth understanding of the field, and we wish to address your concern from the following perspectives.
>
> * **Comparison with related work.** We would first like to respectfully clarify that **all the prior works you mentioned have already been discussed in the Related Work section of our original submission** (Lines 141–147). These modulation-based methods were primarily designed to inject coarse and discrete conditioning signals (e.g., style categories or class labels) uniformly across all tokens. In contrast, our approach is **not** a straightforward or incremental adaptation of these methods, but instead **differs in two essential aspects**:
>
>   * **Token-wise modulation.** Unlike prior works, which apply a global modulation, our approach introduces a token-wise modulation mechanism. Specifically, in prior works, the same conditioning signal is **uniformly applied to all visual patches**, which may suffice for class/style-conditional generation tasks. However, such uniformity falls short for vision-language understanding tasks, which demand fine-grained cross-modal interaction. In contrast, LaVi introduces a novel token-wise modulation strategy, where **customized visual deltas** are dynamically generated for each individual language token. To the best of our knowledge, this is the **first attempt** to bring token-level granularity into feature modulation for multimodal alignment. To validate its necessity, we implemented a baseline that uses **global AdaIN-style modulation** under the same training setup. The results are summarized below:
>     | Model| MMB | SEED-I | TextVQA | DocVQA | ChartQA | AI2D | InfoVQA |
>     |-|-|-|-|-|-|-|-|
>     | w/ AdaIN| 68.4 | 64.7 | 61.9 | 62.4 | 59.7 | 60.3 | 44.5
>     | w/ Ours |  79.3 | 76.9 | 77.0 | 87.6 | 81.3 | 80.9 | 67.5 |
>
>      The results confirm that token-wise modulation leads to significantly better performance, validating the effectiveness and irreplaceability of our proposed method for LVLMs.
>
>
>   * **Multi-conditional modulation support.** Previous works only supports **a single** conditioning input. In contrast, our method allows **multiple image-conditioned signals** to be composed and applied to different parts of the text sequence. This enables LaVi to not only handle multi-tile high-resolution images and videos, but also support more complex interleaved vision–language understanding. To validate this capability, we report LaVi’s performance on five standard multi-image benchmarks:
>
>     | Model| LLaVA-Interleave | MuirBench | Mantis | BLINK | TR-VQA | Avg|
>     |-|-|-|-|-|-|-|
>     |GPT-4V (V-Preview) | 60.3 |62.3 |62.7|51.1| 54.5| 58.2 |
>     | LLaVA-OV-7B | 64.2| 41.8 | 64.2 | 48.2 | 80.1 | 59.7 |
>     | LaVi-7B | 65.6 | 43.7 | 63.5 | 46.9  | 81.8 | 60.3 |
>
>      LaVi achieves comparable or superior performance relative to strong baselines across all five benchmarks. These results demonstrate that our proposed enhancements effectively extend the capabilities of conditional normalization techniques.
>
>   Overall, LaVi represents **the first LVLM built upon a feature-modulation paradigm**, illustrating a promising and feasible architectural direction.
>
> * **Comparison with other VL fusion works.**  Thank you for your rigorous and thoughtful feedback. We provide a **principled analysis** of the underlying mechanism by which feature modulation enables vision–language fusion in **Rebuttal by Authors [4/5]**. Furthermore, in **Rebuttal by Authors [3/5]**, we present a comparative evaluation against diverse fusion architectures. If you believe any additional experiments or analyses would help clarify this aspect further, we would be more than happy to conduct and include them. We sincerely appreciate your constructive suggestions.

---

> ### Author Response · Authors · 2025-11-22
> **Rebuttal by Authors [2/5]**
>
> **W2. Performance Comparison**
>
> * **Performance clarification.** We truly appreciate your rigorous review and **have refined the original manuscript to reflect your suggestions**. Specifically, we believe that LaVi achieves **comparable or improved** performance over the baselines while reducing obvious computational cost. We sincerely hope that our clarifications could earn your valuable understanding and recognition.
>
> * **Additional evaluation benchmarks.** We greatly appreciate your emphasis on comprehensive evaluation. We would like to respectfully clarify that **the benchmark selection in our original submission was aligned with prior work, particularly LLaVA, to ensure fair and direct comparison.** That said, we fully agree that **broader evaluation is essential**. In response to your valuable suggestions, we have now added results on additional benchmarks as detailed below:
>   * **Fine-grained visual understanding.** To more thoroughly assess fine-grained visual understanding, **in addition to TextVQA, we extend our evaluation to benchmarks such as DocVQA, ChartQA, AI2D, OCRBench, and InfoVQA**, which require detailed reasoning over figures, documents, and textual content. The results are summarized in the table below:
>
>     | Model | FLOPs | TextVQA | DocVQA | ChartQA | AI2D | InfoVQA | OCRBench | Avg |
>     | -|-|-|-|-|-|-|-|-|
>     | LLaVA-1.5| 8.4T |58.2 | 23.8 | 17.9 | 52.6 | 21.7 | 20.1 | 32.4 |
>     | LaVi-Image| 0.6T| 58.4 | 24.5 | 17.3 | 52.8 | 21.6 | 21.0 | 32.6 |
>     ||
>     | LLaVA-1.6 | 32.9T| 64.9 | 66.9 | 54.2 | 64.6 | 30.2 | 50.3 | 55.2 |
>     |LaVi-Image (HD) | 1.7T |64.3 | 66.3 | 55.4 | 65.3 | 31.4 | 51.0 | 55.6 |
>     ||
>     | LLaVA-OV-7B |60.4T| 76.1 | 87.3 | 80.3 | 81.4 | 66.3 | 62.7 | 75.7 |
>     | LaVi |3.6T|77.0 | 87.6 | 81.3 | 80.9 | 67.5 | 63.4 | 76.3 |
>
>     The results suggest that LaVi achieves competitive performance in fine-grained visual understanding.
>
>   * **STEM domain.** To more thoroughly assess reasoning capabilities in the STEM domain, we extend our evaluation to benchmarks such as **MMMU, MathVista, and MathVerse**. The results are summarized below:
>
>     | Model | FLOPs| MMMU | MathVista | MathVerse | Avg |
>     | -|-|-|-|-|-|
>     | LLaVA-1.5| 8.4T | 35.6 | 25.5 | 8.0 | 23.0 |
>     | LaVi-Image| 0.6T |35.6 | 26.1 | 8.5 | 23.4 |
>     ||
>     | LLaVA-1.6 | 32.9T| 37.0 | 32.9 | 15.2 | 28.4 |
>     |LaVi-Image (HD) | 1.7T | 36.7 | 33.4 | 16.5 | 28.9 |
>     ||
>     | LLaVA-OV-7B | 60.4T |48.6 |  63.3 | 26.4 | 46.1 |
>     | LaVi |3.6T| 48.8 | 64.2 |  27.2 | 46.7 |
>
>     The results suggest that LaVi achieves competitive performance on STEM reasoning tasks.
>
>   * **Long Video Evaluation.** We would like to respectfully clarify that **our original evaluation on Video-MME reports the overall average score, rather than the performance on the long subset**. However, we agree that specifically evaluating long-form video understanding is meaningful. To address this, we have supplemented our results with evaluations on **Video-MME-Long and LongVideoBench**, as summarized below:
>
>     | Model |FLOPs| Video-MME-Long | LongVideoBench |
>     | -|-|-|-|
>     |VideoLLaVA| 32.6T| 36.2 |39.1|
>     | ShareGPT4Video| 39.2T| 35.0 | 39.7 |
>     | LLaVA-OV | 129.6T| 48.5| 56.3 |
>     | LaVi |18.6T| 49.6 | 56.6 |
>
>     The results demonstrate that LaVi exhibits competitive capabilities in long-form video understanding.

---

> ### Author Response · Authors · 2025-11-22
> **Rebuttal by Authors [3/5]**
>
> **W3. Baselines Comparison**
>
> * **SOTA Models Comparison.** Thank you for raising this important point. We wish to clarify it from two perspectives:
>     * **Choice of baseline models.** Our original comparison primarily focused on LLaVA-series models under similar settings. Following your valuable suggestion, **we have revised the manuscript to avoid potential ambiguity**. We now emphasize that LaVi aims to deliver **comparable performance under significantly reduced computational cost**, rather than to claim **SOTA**.
>
>     * **Verified potential of LaVi.** We also explored whether LaVi can achieve stronger performance with larger-scale data and more powerful LLMs. To this end, we developed a new version using **open-source datasets and backbones**. Specifically, we pre-trained on 85M image-text pairs and fine-tuned on 22M instruction samples from **LLaVA-OneVision-1.5**. The backbone was upgraded to **Qwen2.5-7B**, and the full training process was completed on 64×A100 GPUs in 78 hours. We evaluate performance using LMMs-Eval, and compute FLOPs using the DeepSpeed FLOPs Profiler:
>
>       |Model|FLOPs| Data| Latency | MMB| MMMU| MME| SEED| GQA| TextVQA| DocVQA| ChartQA |InfoVQA| AI2D| RQA| MMStar|Avg|
>       |-|-|-|-|-|-|-|-|-|-|-|-|-|-|-|-|-|
>       |Qwen2-VL-7B| 10.0T|≥1.4T | 242.5ms| 79.1| 50.3| 82.4 |75.9| 62.3| 81.2| 91.6| 81.8| 69.7| 80.3 |65.9| 57.3| 73.1|
>       |Qwen2.5-VL-7B| 19.2T| 4.1T| 333.4ms| 83.8|51.0 |82.2| 77.6| 60.8 |82.9| 94.7| 83.8| 80.1|82.6 |69.9| 62.4| 76.0|
>       |LaVi-7B| 3.8T| ~0.2T |139.7ms| 82.0| 50.6| 82.4 | 76.8| 63.1| 80.6 | 92.2| 82.7| 76.5 |81.3| 67.7| 60.6 |74.7|
>
>       We would like to respectfully emphasize that **the powerful open-source models you mentioned were trained at a significantly larger scale compared to LaVi, e.g., 4.1T token for Qwen2.5-VL-7B vs 0.2T for LaVi-7B**. That said,  LaVi attains even better average performance using only 38% of the FLOPs and 15% of the training tokens required by Qwen2-VL, indicating that FMI offers a highly effective and efficient mechanism for building an LVLM.
>
> * **Diverse Architectures Comparison.** Thank you for your suggestion. We appreciate your broad understanding of the field. We agree that mPLUG-Owl3, VideoChat-Flash, and CrossLMM are all valuable contributions. However, we would like to respectfully clarify that **mPLUG-Owl3 and LaVi are designed as unified image–video understanding models, whereas CrossLMM and VideoChat-Flash focus on video understanding**.
>
>
>     | Model |  LLM | ViT | Image/Video FLOPs | MMB | TextVQA | VideoMME | LongVideoBench  |
>     |-|-|-|-|-|-|-|-|
>     | mPLUG-Owl3 | Qwen2-7B |  Siglip |  4.9/21.2 |77.6 | 69.0 | 53.5 | 52.1 |
>     | VideoChat-Flash | Qwen2-7B | UMT | 4.2/44.8| 72.5 | 66.4 | 69.7 |  64.7|
>     | CrossLMM | Qwen2.5-7B | Siglip2 | --/-- | -- | -- | 62.6| 56.0 |
>     | LaVi | Qwen2-7B |  Siglip |  3.6/18.6  | 79.3 | 77.0 | 57.3 | 56.6 |
>     | LaVi w/ LLaVA-Video-178K| Qwen2-7B |  Siglip | 3.6/18.6 | 80.1 | 77.2 |63.0 | 58.4 |
>
>     We compare performance and FLOPs cost across two image benchmarks and two video benchmarks. For CrossLMM, since we are currently unable to obtain the checkpoint, we regret that we were unable to measure the FLOPs of this excellent model on evaluated benchmarks. Considering that both VideoChat-Flash and CrossLMM are specialized video models trained with approximately 2M video SFT samples, we additionally report the results of LaVi trained on LLaVA-Video-178K for a fair comparison. Overall, the results show that LaVi achieves a competitive balance between computation and accuracy relative to the relevant architectures you mentioned, and its video performance benefits further from additional video data.

---

> ### Author Response · Authors · 2025-11-22
> **Rebuttal by Authors [4/5]**
>
> **W4. Justification and analysis**
>   * **Conditioning modules ablation.**  The three modules share the common purpose of aggregating visual semantics relevant to each text token to generate its modulation parameters, thereby enabling token-wise modulation. We would first like to respectfully emphasize that **Table 4 of the manuscript already compares models trained with the three conditioning modules** (MLP-, Conv-, and Attention-based). The comparison includes **seven benchmark accuracies, training time, and inference FLOPs**, providing an initial view of their efficiency–performance trade-offs. In addition, Appendix B provides the complete code implementations of all three conditioning modules. To provide a clearer comparison, we report **more efficiency metrics** and additionally include their **average accuracy across the all 9 benchmarks in Table 1 of the manuscript** as a reference.
>
>     | Module| Train Time | Train Peak Mem | Infer FLOPs | Infer Latency  | Acc |
>     |-|-|-|-|-|-|
>     | LaVi w/ MLP-based  | 5.84h |  74.7G | 0.83T | 120.7ms | 69.2 |
>     | LaVi w/ Conv-based|  5.95h | 73.6G | 0.76T | 118.5ms | 68.6
>     | LaVi w/ Attention-based| 6.57h | 76.2G | 0.92T |  122.0ms | 70.0 |
>
>     Given that the attention-based conditioning module **achieves higher accuracy under comparable computational cost and the architectural consistency with the underlying LLM**, we adopt the attention-based design as the default choice in our main experiments.
>
>
>    * **Vision-language integration**. We wish to respectfully argue that this phenomenon reveals an essential principle behind visual modulation: the model concentrates visual information on the semantic anchor points within the text sequence that are most critical for downstream decisions. Specifically, the cosine distance before and after injection reflects **how strongly each text token is influenced by visual signals**. Within a sentence, verbs and nouns typically carry richer semantic content than conjunctions or punctuation, making them more likely to align with visual information. When a text token has relevance to the visual content, it naturally receives stronger modulation influences, refining the semantic **specificity and details** of this text token **within the current visual context**. Therefore, Figure 7 illustrates the mechanism by which LaVi integrates vision and language at the token level: visual signals are used to refine the semantic representations of relevant content words, anchoring them precisely to the current visual context, while minimally altering function words that primarily serve syntactic roles and exhibit weak visual relevance.

---

> ### Author Response · Authors · 2025-11-22
> **Rebuttal by Authors [5/5]**
>
> **W5. Contribution Discussion**.
>
> Thank you for your careful and rigorous feedback. We would like to address your concerns from two perspectives:
>
> * **Manuscript revision.** We have carefully revised the manuscript to correct several expressions that may have caused potential misunderstandings about the contribution of LaVi.
>
> * **Contribution clarification.** We would like to respectfully emphasize that, built upon the proposed modulation-based paradigm, LaVi has the potential to simultaneously achieve **top performance and efficiency**, rather than merely representing an engineering trade-off between the two. As elaborated in **Rebuttal by Authors [3/5]**, LaVi attains performance **comparable to the leading models** you highlighted, while using less than **30%** of the computational cost and fewer than **15%** of the training tokens. We will also **release this version** to support reproducibility and foster future research.
>
> We fully understand and respect your high standards for rigor and verification, and we have truly made every possible effort within our capacity. We sincerely hope to earn your understanding and recognition. We would be deeply grateful for your consideration. If you require any further clarification or additional experiments, we would be more than willing to provide them.

---

> ### Author Response · Authors · 2025-11-27
> **Official Comment by Authors**
>
> Dear Reviewer u2BS,
>
> Thank you for your constructive feedback and for the opportunity to strengthen our work. In response to your valuable comments, we have uploaded a revised version and provided detailed point-by-point answers in the rebuttal. We would greatly appreciate it if you could take a look.
>
> If you have any additional suggestions or questions, we would be happy to discuss them and further refine the paper.
>
> Thank you again for your time and feedback😄😄!

---

### Official Review · Reviewer_tgaP · 2025-11-02

**Soundness:** 3
**Presentation:** 3
**Contribution:** 2
**Rating:** 4
**Confidence:** 4

**Summary:**

The manuscript focuses on the vision-language interation in the paradigm of vision-language models (VLMs). Existing strategies for integrating vision information in VLMs either introduces extra learnable parameters within the language model, disrupting architecture consistency, or concatenates vision tokens with language tokens, leading to reduced efficiency. To achieve minimal structural interference and computational scalability at the same time, the manuscript presents a new method LaVi. The core mechanism is to modulate the features through modulating the LayerNorm parameters in the language models, conditioned on vision features.

Experiment shows reduced FLOPs and inference latency, while keeping a competitive performance compared to token concatenation baselines. Ablation studies are conducted on different modulated components, modulation parameters, and modulation frequency.

**Strengths:**

1. The approach is well-motivated to improve the efficiency of vision-language models by altering the way of injecting vision information into language models.

2. The overall idea and implementation of feature modulation injection is simple.

3. Empirical results show that LaVi is highly-efficient, using less than 10% computation, 50% latency to achieve a similar performance to token concatenation baselines.

4. Ablation studies are conducted to show the effectiveness of the proposed modules.

**Weaknesses:**

1. One of the major flaw of the proposed feature modulation injection is that, similar to the adaLN in diffusion transformers, support only one set of visual input. For multiple visual inputs (multiple input images, not multiple frames or image tiles mentioned in the manuscript), the difficulty lies in choosing which visual input to condition the layernorm. This limits the application of the proposed feature modulation injection to broader applications of VLMs, hence hardly able to be employed in existing VLMs.

2. The experiment mainly focus on general visual understanding benchmarks, while the main advantage of concatenating visual tokens is that it can preserve enough details for fine-grained recognition, such as OCR. The manuscript fail to evaluate the performance of feature modulation injection on such benchmarks. This is especially important when we see the performance advantage diminishes on TextVQA when the resolution of input images increase (LaVi-Image vs LLaVA-v1.5, and LaVi-Image (HD) vs LLaVA-v1.6).

3. What is the advantage of not modifying the architecture of the language model? Does it better preserve the language capability? This is not reflected in the manuscript.

4. The experiment focuses on 7B language model. It is unknown how the method scales with different number of parameters in the language model.

**Questions:**

1. The feature modulation injection seems highly similar to adaLN in diffusion tranformers. It would be better if explanations of differences are provided.

---

> ### Author Response · Authors · 2025-11-22
> **Rebuttal by Authors [1/5]**
>
> Dear Reviewer tgaP,
>
> We sincerely appreciate your valuable and insightful comments. We found them extremely helpful for improving our manuscript. We will strive to address each comment in detail, one by one below.
>
> ---
>
> **W1. Support for multi-image understanding**
>
> This is indeed a very insightful question, and we fully agree that multi-image understanding represents an important frontier for advancing LVLM capabilities. We would like to respectfully clarify that **the proposed Feature Modulation Injection (FMI) can naturally accommodate multiple visual inputs**. We wish to address this concern through the following three points:
>
>  * **How FMI supports multi-image understanding.** First, analogous to how LaVi distinguishes frames in video inputs using frame embeddings, multi-image inputs are firstly handled by assigning an **image-level embedding** to all patch tokens belonging to the same image. Distinct embeddings across images allow the conditioning module to differentiate among them. Furthermore, multi-image tasks are typically composed of two basic forms and their combinations.
>
>     * **Joint understanding over multiple images** (e.g., describing similarities, differences, or changes across images), where the input typically follows the format $[\text{IMG}_1,...,\text{IMG}_N, \text{Text}]$, In this case, distinguishing images using the image-level embedding is sufficient for effective conditioning.
>
>     * **Interleaved image–text understanding** (e.g., visual storytelling), where the input may take the form  $[\text{IMG}_1,\text{Text}_1,\text{IMG}_2,\text{Text}_2,...]$. For such settings, we incorporate **causality** into the conditioning module. The tokens in $\text{Text}_i$ are modulated only by the visual features of the preceding images $[\text{IMG}_1,...,\text{IMG}_i]$. Different images are also distinguished by image-level embedding.
>
>    Based on these principles, **the processing of any multi-image input can be unified as follows**: all text segments $\text{Text}_i$ are concatenated and fed into the LLM, while all images $\text{IMG}_i$ are encoded by the ViT and concatenated in their original order.  In conditioning module, each token in $\text{Text}_i$ constructs its visual conditioning by aggregating information from all images that precede it in the original sequence, enforced through a **causal mask**. It allows LaVi to seamlessly support the multi-image training data used in LLaVA-OneVision-Instruct.
>
>
> * **How FMI performs on multi-image understanding**: Following the evaluation protocol of LLaVA-OneVision, we assess the multi-image capability of LaVi on five established multi-image benchmarks, using LMMs-Eval as the evaluation toolkit. The results are summarized below:
>
>     | Model| LLaVA-Interleave | MuirBench | Mantis | BLINK | TR-VQA | Avg|
>     |-|-|-|-|-|-|-|
>     |GPT-4V (V-Preview) | 60.3 |62.3 |62.7|51.1| 54.5| 58.2 |
>     | LLaVA-OV-7B | 64.2| 41.8 | 64.2 | 48.2 | 80.1 | 59.7 |
>     | LaVi-7B | 65.6 | 43.7 | 63.5 | 46.9  | 81.8 | 60.3 |
>
>     The results indicate that LaVi attains superior or comparable performance to LLaVA-OV-7B, demonstrating the effective support for multi-image understanding.
>
> * **Related results are included in the paper**: We originally did not discuss multi-image performance in the main text because several baseline models (e.g., LLaVA v1.5 and LLaVA v1.6) do not evaluate multi-image inputs, making direct comparisons less informative. Following your valuable suggestion, we have now added the corresponding multi-image experiments to **Appendix C.2**. We sincerely thank you once again for this helpful feedback, which has improved the completeness of LaVi.

---

> ### Author Response · Authors · 2025-11-22
> **Rebuttal by Authors [2/5]**
>
> **W2. Evaluation on fine-grained visual understanding**
>
> Thank you for your valuable suggestions. We respectfully emphasize that **LaVi possesses satisfactory fine-grained visual understanding capability** through precise token-wise modulation, and we wish to address your concern from the following perspective:
>
> * **More comprehensive evaluation.** Our initial choice of benchmarks was primarily driven by comparability with the LLaVA series, and thus focused on those reported in the original papers. To more thoroughly assess fine-grained visual understanding, **in addition to TextVQA, we extend our evaluation to benchmarks such as DocVQA, ChartQA, AI2D, OCRBench, and InfoVQA**, which require detailed reasoning over figures, documents, and textual content. The results are summarized in the table below:
>
>     | Model | FLOPs | TextVQA | DocVQA | ChartQA | AI2D | InfoVQA | OCRBench | Avg |
>     |-|-|-|-|-|-|-|-|-|
>     | LLaVA-1.5| 8.4T | 58.2 | 23.8 | 17.9 | 52.6 | 21.7 | 20.1 | 32.4 |
>     | LaVi-Image| 0.6T | 58.4 | 24.5 | 17.3 | 52.8 | 21.6 | 21.0 | 32.6 |
>     ||
>     | LLaVA-1.6 | 32.9T | 64.9 | 66.9 | 54.2 | 64.6 | 30.2 | 50.3 | 55.2 |
>     |LaVi-Image (HD) | 1.7T |64.3 | 66.3 | 55.4 | 65.3 | 31.4 | 51.0 | 55.6 |
>     ||
>     | LLaVA-OV-7B | 60.4T |76.1 | 87.3 | 80.3 | 81.4 | 66.3 | 62.7 | 75.7 |
>     | LaVi |3.6T | 77.0 | 87.6 | 81.3 | 80.9 | 67.5 | 63.4 | 76.3 |
>
>   The results suggest that LaVi is not inferior to visual token concatenation in terms of fine-grained visual understanding. We further discuss LaVi’s advantages in visual scalability in the next point.
>
> * **High efficiency enables greater scalability.**  We would also like to respectfully clarify an important point: fine-grained recognition often relies on representing visual content with a larger number of tokens. However, for traditional approaches, extending the length of visual token sequences comes at a substantial computational cost (e.g., FLOPs increase from 8.4T in LLaVA-v1.5 to 32.9T in LLaVA-v1.6, and further to 60.4T in LLaVA-OV). In contrast, under the same visual token scaling strategy, LaVi's computational cost increases only modestly (e.g., from 0.6T to 1.7T and then to 3.6T). It indicates that **LaVi can further enhance the granularity of visual inputs while maintaining low computational overhead**. Specifically, we train and then evaluate an **extreme case** where every input image is divided into a 4×4 grid of tiles for LaVi. The corresponding results are summarized below:
>
>     | Model | FLOPs | HighRes Strategy | TextVQA | DocVQA | ChartQA | AI2D | InfoVQA | OCRBench | Avg |
>     |-|-|-|-|-|-|-|-|-|-|
>      | LLaVA-OV-7B | 60.4T |  AnyRes-Max9 | 76.1 | 87.3 | 80.3 | 81.4 | 66.3 | 62.7 | 75.7 |
>     | LaVi | 3.6T |  AnyRes-Max9 | 77.0 | 87.6 | 81.3 | 80.9 | 67.5 | 63.4 | 76.3 |
>     | LaVi | 19.5T |  AnyRes-16 | 77.8 | 88.2 | 81.6 | 81.8 | 68.0 | 64.3| 77.0 |
>
>   These results demonstrate that LaVi offers a significant **efficiency advantage** when scaling up the visual sequence length to enable more fine-grained understanding of input images.
>
> * **Related results are included in the paper.** We have incorporated the above analysis and findings into **Appendix C.1**. We are sincerely grateful for your valuable feedback, which has made a significant contribution to improving the quality of our manuscript.

---

> ### Author Response · Authors · 2025-11-22
> **Rebuttal by Authors [3/5]**
>
> **W3. Preservation of language capability**
>
> As you insightfully pointed out, we also believe that preserving the architecture of the language model plays a crucial role in maintaining language capability. We would like to respectfully highlight that **this aspect has already been discussed in Table 4 and Figure 5 of our original submission**. Specifically:
>
>
> * **Comparison across injection paradigms.** As shown in Table 4, we compare the proposed Feature Modulation Injection (FMI) with two representative paradigms, In-context Injection (e.g., LLaVA) and Architectural Injection (e.g., Flamingo), under exactly the same experimental settings. When evaluated on language benchmarks such as MMLU, MBPP, and MATH, LaVi achieves accuracy that remains close to the original LLM while outperforming existing LVLMs, indicating stronger preservation of language capability.
>
> * **Layer-wise similarity to the base LLM.** As illustrated in Figure 5, we feed MMLU samples into models trained with each injection paradigm and compute the layer-wise cosine distance between their hidden states and those of the original LLM. The results show that FMI yields consistently smaller distances across layers, which aligns with stronger performance. This supports our claim that FMI better preserves the pretrained language representations.
>
> * **Open to any further analysis if helpful.** If you believe any additional experiments or analyses would help clarify this aspect further, we would be more than happy to conduct and include them. We sincerely appreciate your thoughtful feedback.

---

> ### Author Response · Authors · 2025-11-22
> **Rebuttal by Authors [4/5]**
>
> **W4. Scaling with LLM size**
>
> Thank you for your valuable suggestion. To explore the scalability of our method across different model sizes, we evaluate LaVi using three variants of Qwen2 with different parameter scales: 0.5B, 7B, and 72B. For the 0.5B and 72B models, we follow the same training setup and data as used for LaVi-7B. Benefiting from the computational efficiency of LaVi, even the 72B variant can be trained within 2 days on 32×A100 GPUs.
>
> The image-based performance is summarized as follows:
>
> | Model| MMB | SEED-I | TextVQA | DocVQA | ChartQA | AI2D | InfoVQA |
> |-|-|-|-|-|-|-|-|
> | LaVi-0.5B | 51.6 | 67.4 | 65.9 | 69.5 | 61.2 | 57.4 | 43.1 |
> | LaVi-7B |  79.3 | 76.9 | 77.0 | 87.6 | 81.3 | 80.9 | 67.5 |
> | LaVi-72B | 85.7 | 77.8 | 80.5 | 93.8 | 84.7 | 85.7 | 75.6 |
>
> The video-based performance is summarized as follows:
>
> | Model| Frame | EgoSchema | MLVU | VideoMME | MVBench |
> |-|-|-|-|-|-|
> | LaVi-0.5B | 32 | 28.7 |  51.6 | 45.3 | 44.9 |
> | LaVi-7B | 32 | 58.4 | 62.3 |  57.3 | 56.5 | 58.6 |
> | LaVi-72B | 32 |61.8 | 68.5 | 64.1 | 60.2 |
>
> The results demonstrate that **LaVi scales favorably across model sizes**, with consistent performance improvements observed from 0.5B to 72B on both image and video benchmarks.

---

> ### Author Response · Authors · 2025-11-22
> **Rebuttal by Authors [5/5]**
>
> **Q1. Difference with AdaLN**
>
> We appreciate your deep understanding of foundational architectures in vision-language models. We wish to address your question from the following two perspectives.
>
> * **Token-wise modulation.** Unlike AdaLN, which applies a global modulation, our approach introduces a token-wise modulation mechanism. Specifically, in DiT, the same conditioning signal is **uniformly applied to all visual patches**, which may suffice for class-conditional generation tasks. However, such uniformity falls short for vision-language understanding tasks, which demand fine-grained cross-modal interaction. In contrast, LaVi introduces a novel token-wise modulation strategy, where **customized visual deltas** are dynamically generated for each individual language token. To the best of our knowledge, this is the **first attempt** to bring token-level granularity into feature modulation for multimodal alignment. To validate its necessity, we implemented a baseline that uses **global AdaLN-style modulation** under the same training setup. The results are summarized below:
>     | Model| MMB | SEED-I | TextVQA | DocVQA | ChartQA | AI2D | InfoVQA |
>     |-|-|-|-|-|-|-|-|
>     | w/ AdaLN| 68.4 | 64.7 | 61.9 | 62.4 | 59.7 | 60.3 | 44.5
>     | w/ Ours |  79.3 | 76.9 | 77.0 | 87.6 | 81.3 | 80.9 | 67.5 |
>
>   The results confirm that token-wise modulation leads to significantly better performance, validating the effectiveness and irreplaceability of our proposed method for LVLMs.
>
> * **Multi-conditional modulation support.** As you noted in W1, AdaLN supports **only a single** conditioning input. In contrast, our method allows **multiple image-conditioned signals** to be composed and applied to different parts of the text sequence. This enables LaVi to not only handle multi-tile high-resolution images and videos, but also support more complex interleaved vision–language understanding.
>
> Overall, LaVi represents **the first LVLM built upon a feature-modulation paradigm**, illustrating a promising and feasible architectural direction.

---

> ### Author Response · Authors · 2025-11-27
> **Official Comment by Authors**
>
> Dear Reviewer tgaP,
>
> Thank you for your constructive feedback and for the opportunity to strengthen our work. In response to your valuable comments, we have uploaded a revised version and provided detailed point-by-point answers in the rebuttal. We would greatly appreciate it if you could take a look.
>
> If you have any additional suggestions or questions, we would be happy to discuss them and further refine the paper.
>
> Thank you again for your time and feedback😄😄!

---

### Official Review · Reviewer_bSwR · 2025-11-02

**Soundness:** 3
**Presentation:** 3
**Contribution:** 3
**Rating:** 6
**Confidence:** 4

**Summary:**

This paper proposes a novel vision-language model ensemble method, termed Language and Vision Integrator (LaVI), which injects visual information into the affine parameters of Layer Normalization through Vision-Infused Layer Normalization (ViLN), avoiding the context expansion problem of traditional methods. This method significantly improves computational efficiency while maintaining performance.

**Strengths:**

1. LaVI uses layer-normalized affine parameters, avoiding complexity issues caused by excessively long contexts.

2. LaVI achieves significant efficiency improvements.

**Weaknesses:**

1. Lack of necessary theoretical analysis (detailed in questions).

2. Lack of comparison with some existing work:

[a] Shaolei Zhang, Qingkai Fang, Zhe Yang, Yang Feng. LLaVA-Mini: Efficient Image and Video Large Multimodal Models with One Vision Token.

[b] Bo Tong, Bokai Lai, Yiyi Zhou, Gen Luo, Yunhang Shen, Ke Li, Xiaoshuai Sun, Rongrong Ji. FlashSloth: Lightning Multimodal Large Language Models via Embedded Visual Compression.

**Questions:**

1. Why can effective visual-language alignment be achieved through affine parameter modulation? There is a lack of in-depth theoretical analysis.

2. Specific sample analyses can be performed to demonstrate the visual perception of LaVI. For example, to specifically describe a complex scene.

3. How does the model perform on fine-grained tasks, such as OCR-related tasks like TextVQA[c], DocVQA[d]?

[c] Singh, Amanpreet and Natarjan, Vivek and Shah, Meet and Jiang, Yu and Chen, Xinlei and Parikh, Devi and Rohrbach, Marcus. Towards VQA Models That Can Read.

[d] Minesh Mathew, Dimosthenis Karatzas, C.V. Jawahar. DocVQA: A Dataset for VQA on Document Images.

---

> ### Author Response · Authors · 2025-11-22
> **Rebuttal by Authors [1/4]**
>
> Dear Reviewer bSwR,
>
> We sincerely appreciate your valuable and insightful comments. We found them extremely helpful for improving our manuscript. We will strive to address each comment in detail, one by one below.
>
> ---
>
> **W1 & Q1. Theoretical analysis**
>
> Thank you for your thoughtful and detailed consideration. We wish to address your question from the following two perspectives:
>
> * **What does affine parameter modulation do?** We argue that the visual-conditioned modulation mechanism fundamentally implements a form of domain adaptation [1–3]. In this view, the source domain ($\mathcal{D} _ \text{src}$) corresponds to the general language distribution acquired by the LLM during pretraining, while the target domain ($\mathcal{D} _ \text{tgt}$) corresponds to the conditional language distribution induced by vision–language joint modeling. The injection of visual signals inevitably introduces a domain shift. Domain adaptation therefore seeks to minimize the divergence between $\mathcal{D} _ {\text{src}}$ and $\mathcal{D} _ {\text{tgt}}$, enabling the newly introduced visual semantics to be absorbed into the pretrained source-domain distribution, thereby achieving **vision–language alignment**. The divergence between the source and target domain is often approximated by moment matching [3,4]:
>
>   $$
>      \mathcal{D}(\mathcal{P} _ {\text{src}}, \mathcal{P} _ {\text{tgt}}) \approx \sum_{k=1}^{K} \left| \mathbb{E} _ {h \sim \mathcal{P}{\text{src}}}[h^{\otimes k}] - \mathbb{E}_{h \sim \mathcal{P}{\text{tgt}}}[h^{\otimes k}] \right|^2
>   $$
>      Here, $h^{\otimes k}$ denotes the $k$-th order moment of the hidden representation $h$. In practice, $k = 2$ suffices: the first moment (mean $\mu$) determines where the representation lies in the semantic space, and the second moment (variance $\sigma$) determines how strongly each semantic direction is expressed. This theoretical insight directly motivates our design. **Affine parameter modulation offers a precise and direct interface for manipulating these moments**:
>     $$
>       \text{ViLN}(t, v) = (\alpha + \Delta{\alpha_v}) \odot t + (\beta + \Delta{\beta_v})
>     $$
>     where the first-order moment of the hidden representation is controlled by the shift term $(\beta + \Delta{\beta_v})$, and the second-order moment is controlled by the scale term $(\alpha + \Delta{\alpha_v})$. This direct control means that the learning of the visual deltas is functionally equivalent to optimizing the distributional divergence, ensuring the transformation is minimal yet sufficient to project visual concepts into the LLM’s native linguistic space.
>
> * **What does affine parameter modulation achieve?** Based on the above theory, we assert that modulating the mean and variance learns an efficient projection of visual concepts onto the LLM's native linguistic space. We provide an empirical verification of this claim. Specifically, we randomly sampled **2k examples** from all benchmarks reported in Table 1. For each example, we fed the **image–question pair** into LaVi and LLaVA-OV and extracted the hidden states of the question tokens. We then fed the **question alone** into the shared base LLM (Qwen2-7B) and collected the corresponding hidden states. Pairwise cosine distances among the three models were computed every 7 layers. The results are shown below:
>
>   | Relation | Layer 0 | Layer 7 | layer 14 | Layer 21 | Layer 28 |
>   |-|-|-|-|-|-|
>   | LLaVA-OV vs Qwen2| 0.00 | 0.12 | 0.15 | 0.19 | 0.22 |
>   | LaVi vs Qwen2 | 0.00 | 0.05 | 0.07 | 0.13 | 0.17 |
>   | LLaVA-OV vs LaVi | 0.00 | 0.03 | 0.03 | 0.04 | 0.05 |
>
>   These results clearly show that visual injection shifts the hidden states away from those of the base LLM. However, since LaVi achieves domain adaptation via minimal statistical rectification, its features remain closer to the base LLM  than those of LLaVA-OV. More importantly, the hidden state distances between LaVi and LLaVA-OV remain relatively small across layers. This indicates that modulation achieves a **similar visual–language alignment effect** to that of the in-context injection, while incurring substantially lower computational overhead.
>
>  If you believe any additional experiments or analyses would help clarify this aspect further, we would be more than happy to conduct and include them. We sincerely appreciate your thoughtful feedback.
>
>
> [1] Ben-David S, Blitzer J, Crammer K, et al. A theory of learning from different domains[J]. Machine learning, 2010, 79(1): 151-175.
>
> [2] Tzeng E, Hoffman J, Zhang N, et al. Deep domain confusion: Maximizing for domain invariance[J]. arXiv preprint arXiv:1412.3474, 2014.
>
> [3] Long M, Cao Y, Wang J, et al. Learning transferable features with deep adaptation networks[C]//International conference on machine learning. PMLR, 2015: 97-105.
>
> [4] Gretton A, Borgwardt K M, Rasch M J, et al. A kernel two-sample test[J]. The journal of machine learning research, 2012, 13(1): 723-773.

---

> ### Author Response · Authors · 2025-11-22
> **Rebuttal by Authors [2/4]**
>
> **W2. Existing work comparison**
>
> We appreciate your comprehensive insights into the field. We provide a comparison with the two related works below. We follow the performance reported in the original papers and compute FLOPs using the DeepSpeed FLOPs Profiler.
>
> | Model | FLOPs| GQA | TextVQA |  SEED-I | MMB | POPE | SQA | Avg.|
> |-|-|-|-|-|-|-|-|-|
> |LLaVA-OV | 60.4T | 62.2 | 76.1 | 75.4 | 80.8 | 87.4 | 96.0 | 79.7 |
> |LLaVA-Mini| 2.4T | 61.3 | 58.5 | 63.0 | 71.6 | 85.3 | 83.1 | 70.5 |
> |FlashSloth| 4.9T | 62.5 | 71.0 | 71.2 | 75.7 | 87.2 | 91.1 | 76.5 |
> |LaVi| 3.6T |  65.0 | 77.0 | 76.9 | 79.3 | 87.1 | 95.4 | 80.1|
>
> The results show that LaVi achieves better overall performance while maintaining competitive computational efficiency compared to both baselines.

---

> > ### Author Response · Authors · 2025-11-22
> > **Rebuttal by Authors [3/4]**
> >
> > **Q2. Sample analyses**
> >
> > Thank you for your thoughtful consideration. Sample analyses are indeed valuable for strengthening the completeness of our paper. We provide case studies in **three representative scenarios**: fine-grained visual perception, complex chart reasoning, and long-form video understanding. Besides, we provide a visualization of how vision modulation influences the next-token prediction distribution during the LLM decoding process. We respectfully invite you to refer to **Appendix D.CASE STUDY** for further details.

---

> ### Author Response · Authors · 2025-11-22
> **Rebuttal by Authors [4/4]**
>
> **Q3. Fine-grained tasks evaluation**
>
> Thank you for your thoughtful consideration. Our initial choice of benchmarks was primarily driven by comparability with the LLaVA series, and thus focused on those reported in the original papers. To more thoroughly assess fine-grained visual understanding, **in addition to TextVQA, we extend our evaluation to benchmarks such as DocVQA, ChartQA, AI2D, OCRBench, and InfoVQA**, which require detailed reasoning over figures, documents, and textual content. The results are summarized in the table below:
>
> | Model | FLOPs | TextVQA | DocVQA | ChartQA | AI2D | InfoVQA | OCRBench | Avg |
> |-|-|-|-|-|-|-|-|-|
> | LLaVA-1.5| 8.4T | 58.2 | 23.8 | 17.9 | 52.6 | 21.7 | 20.1 | 32.4 |
> | LaVi-Image| 0.6T | 58.4 | 24.5 | 17.3 | 52.8 | 21.6 | 21.0 | 32.6 |
> ||
> | LLaVA-1.6 | 32.9T | 64.9 | 66.9 | 54.2 | 64.6 | 30.2 | 50.3 | 55.2 |
> |LaVi-Image (HD) | 1.7T |64.3 | 66.3 | 55.4 | 65.3 | 31.4 | 51.0 | 55.6 |
> ||
> | LLaVA-OV-7B | 60.4T |76.1 | 87.3 | 80.3 | 81.4 | 66.3 | 62.7 | 75.7 |
> | LaVi |3.6T | 77.0 | 87.6 | 81.3 | 80.9 | 67.5 | 63.4 | 76.3 |
>
> The results suggest that LaVi achieves competitive performance in fine-grained visual understanding.

---

> > ### Comment · Reviewer_bSwR · 2025-11-26
> >
> > Thanks for your response.
> >
> > From the experiments, the proposed method can achieves better performance on fine-grained tasks.

---

> > > ### Author Response · Authors · 2025-11-26
> > > **Official Comment by Authors**
> > >
> > > Dear Reviewer bSwR,
> > >
> > > Thank you for your recognition of our work and the time and effort you have invested as a reviewer!
> > >
> > > We will adhere to your valuable suggestions to refine our manuscript accordingly.

---

### Author Response · Authors · 2025-12-01
**Summary Comment by Authors**

Dear ACs and Reviewers:

We sincerely appreciate the constructive feedback provided by all reviewers and the additional efforts made by the ACs under these special circumstances.

---

### **Summary of Discussion**

During the discussion period, we were fortunate to receive further feedback from 3 reviewers. **All of them acknowledged our responses and decided to maintain or raise their scores to a positive rating.** The details are as follows:

* **Reviewer `5Rmb` (Nov 24, 14:35 AOE) increases the score from 4 to 8.**
    > *[...] Based on these clarifications, I acknowledge the potential of LaVi as an innovative vision-language integration mechanism. I have raised my score to Accept.*

* **Reviewer `bSwR` (Nov 25, 19:20 AOE) increases the score from 6 to 8.**
    > *From the experiments, the proposed method can achieves better performance on fine-grained tasks.*

* **Reviewer `nXfH` (Nov 25, 11:38 AOE) maintains an initial score of 6.**
    > *I thank the authors for the detailed response. The manuscript should be revised accordingly.*

---
### **Summary of Rebuttal**

First, we are encouraged to note that the reviewers provided **highly positive assessments and recognition**:

* **Motivation:** Our method addresses the "***critical problem***" of LVLM computational efficiency (Reviewer `u2BS`) and is considered "***well-motivated***" (Reviewer `tgaP`).

* **Methodology:** The proposed feature modulation is praised as an "***elegant***" (Reviewer `nXfH`), "***creative***" (Reviewer `5Rmb`), and "***concise***" (Reviewers `bSwR`, `tgaP`) design.

* **Performance:** We demonstrate "***strong empirical results and efficiency***" (Reviewers `nXfH`, `5Rmb`, `u2BS`, `tgaP`, `bSwR`). Furthermore, our ablation studies and visualizations are acknowledged as "***comprehensive***" and "***clear***" (Reviewers `nXfH`, `5Rmb`, `tgaP`).

The reviewers raised several insightful questions. We conducted additional experiments and targeted analyses, which can be categorized into three main areas:

**Regarding Experiment**:

* **Comprehensive Evaluation**: Building upon the **initial 18 benchmarks**, we expanded our evaluation to include **20 additional benchmarks**: 5 for fine-grained perception, 5 for multi-image understanding, 2 for caption generation, 4 for complex reasoning, 2 for STEM tasks, and 2 for long-form video understanding. **The results across a total of 38 benchmarks further validate our method for generalizability and competitive performance, while significantly reducing FLOPs by 90% compared to mainstream baselines.**

* **Competitiveness with SOTA**: When empowered by advanced backbones and data, LaVi achieves performance comparable to powerful open-source models such as Qwen2/2.5-VL-7B. **Notably, LaVi surpasses the average performance of Qwen2-VL-7B while requiring only 38% of the FLOPs and 15% of the training tokens.**

* **Model Scaling Laws**: We leverage LLM backbones across three distinct scales: **0.5B, 7B, and 72B**. The results across 11 image and video benchmarks validate the strong parameter scalability.

* **Ablation Studies**: We provide analyses comparing different conditioning modules to justify our default design choice. We conducted ablation studies on parameter initialization strategies. We evaluated the robustness against 3 types of perturbations, demonstrating that LaVi exhibits superior robustness compared to mainstream baselines.


**Regarding Methodology**:

* **Theoretical Analysis**: We provide a theoretical analysis from the perspective of **domain adaptation**. We show that modulating the mean and variance of linguistic features based on visual input is equivalent to minimizing the divergence between the source and target domains via **moment matching**. This mechanism enables visual semantics to be effectively absorbed into the pretrained source-domain distribution.

* **Related Work Discussion**: We conducted a comparative analysis against other related works focusing on efficiency and architectures. **As the first LVLM based on feature modulation, LaVi strikes an excellent balance between performance and efficiency across image and video evaluations.** Besides, we introduce the first modulation scheme that is both token-wise and multi-condition compatible. This design is fundamentally distinct from techniques used in the image generation domain, which is also substantiated by our experimental validation.


**Regarding Visualization**:

* We provide visualizations showing how visual modulation influences the **next-token prediction distribution** during the LLM decoding process. Additionally, we present case studies across **three representative scenarios**: fine-grained visual perception, complex chart reasoning, and long-form video understanding, to highlight the capabilities.

---

Once again, we extend our heartfelt thanks to all reviewers and the ACs for their diligent efforts and significant contributions.

Best regards,

The Authors

---

### Meta-Review · Area_Chair_9kZq · 2025-12-31

**Summary:**

The paper proposes LaVi, a Large Vision-Language Model (LVLM) designed for computational efficiency. The central idea is to inject visual information into a Large Language Model (LLM) by modulating the affine parameters of its internal LayerNorm (LN) modules. Key reviewer concerns: (1) Insufficient novelty: core idea of normalizer parameter modulation is not new; token-wise/multi-conditional extensions are incremental rather than fundamental (u2BS); (2) Unconvincing performance claims: overstated "SOTA" with regressions on key benchmarks (e.g., VQAv2, MMBench), incomplete evaluation of challenging tasks, and outdated baselines (lacking Qwen2-VL/InternVL, u2BS). Authors supplemented the rebuttal with: comparisons to related works (e.g., ATP-LLaVA), expanded evaluations (fine-grained/STEM/long-video tasks), and robustness experiments (noisy/adversarial inputs).  While addressing a meaningful efficiency bottleneck and supplementing experiments in rebuttal, the work fails to resolve core reviewer concerns (like novelty, performance validity, theoretical depth), leading to a Reject decision.

**Reviewer Scores:**

NA

---

### Decision · Program_Chairs · 2026-01-26

Reject